# LEGO-PUZZLES: HOW GOOD ARE MLLMS AT MULTI-STEP SPATIAL REASONING?

## ABSTRACT

Real-world application of spatial intelligence, such as robotic control, autonomous driving, and automated assembly, often require spatial reasoning across multiple sequential steps, yet the extent to which current Multimodal Large Language Models (MLLMs) possess this capability remains largely unexplored. Based on LEGO construction, a recreational activity that critically relies on multi-step spatial reasoning, we introduce **LEGO-Puzzles**, a benchmark designed to systematically evaluate the spatial reasoning capabilities of MLLMs from basic spatial understanding to complex multi-step planning. LEGO-Puzzles contains two task sets. The **Elementary** set covers 11 visual question-answering (VQA) tasks with $1,100$ carefully curated samples to test elementary spatial reasoning skills that are cruical for LEGO assembly. The **Planning** set directly requires the model to generate a step-by-step plan for assembling a target LEGO structure, where the number of intermediate steps required to complete the task varies from 1 to 8. Our evaluation of 23 state-of-the-art MLLMs shows that even the strongest models struggle with elementary reasoning tasks, falling at least $20\%$ behind human performance. The planning accuracy also quickly drops to $0\%$ as the number of steps increases, while our human participants solve all the tasks perfectly. Furthermore, changing the output format of LEGO-Puzzles tasks from multiple choice to image generation significantly reduces performance to near zero. Only GPT-4o and Gemini-2.0-Flash exhibit a limited ability to follow the image generation instructions, while other MLLMs either replicate the input image or generate completely irrelevant outputs. Overall, LEGO-Puzzles reveals critical limitations in current MLLMs' spatial reasoning capabilities and highlights the need for substantial advances.

## 1 INTRODUCTION

Multimodal Large Language Models (MLLMs) have made remarkable progress and have shown promising capabilities in spatial intelligence (Bornstein, 1986). The most elementary capability of spatial intelligence is the ability to perceive and understand spatial relationships among 3D objects from a static scene, which we refer to as **spatial understanding** in this work. This includes many Visual Question Answering (VQA) tasks, such as identifying object attributes (e.g., shape, color, size), recognizing spatial relations (e.g., left/right, above/below), and counting objects that satisfy certain conditions (Johnson et al., 2017; Li et al., 2023b; Wang et al., 2023; Liao et al., 2024).

Many other tasks not only require the spatial understanding of the given scene, but also critically rely on the ability to imagine the consequences of hypothetical actions applied to the scene. That is, one has to sequentially apply the spatial understanding to two or more real or imaginary scenes to solve these tasks, a capability which we refer to as **sequential spatial reasoning** in this work. Notably, a series of benchmarks have been introduced to evaluate the ability of imagining the scene after changing the camera perspective (Shiri et al., 2024; Ma et al., 2024; Jung et al., 2025; Stogiannidis et al., 2025; Li et al., 2025). Ma et al. (2024) further design tasks such as imagining a single step of paper folding.

However, all the above benchmarks focus on only one step of action. In real-world applications, such as robotic control, autonomous driving, and automated assembly, often require performing spatial reasoning and planning for multiple steps. For example, assembling a target structure requires a robotic arm to perceive the current and target states and plan an ordered sequence of actions to

achieve the target state. This not only requires the ability to imagine one step of action, but also the ability to perform spatial reasoning for multiple sequential steps. While this multi-step spatial reasoning is crucial for real-world applications, only a few benchmarks (Yang et al., 2024; Su et al., 2024) contain tasks relevant to this capability, such as path-planning in household.

**Our Contributions.** In this work, we take inspiration from a common recreational activity, LEGO construction, to design an evaluation framework for assessing the spatial intelligence of MLLMs. The assembly process of a complete LEGO model typically encompasses dozens or even hundreds of step-by-step constructions, providing an ideal foundation for testing sequential reasoning abilities. Moreover, LEGO construction has been widely used in cognitive science as a reliable indicator of spatial intelligence. Studies have shown that LEGO construction is strongly associated with core spatial skills such as *mental rotation* and *visuo-spatial working memory*, as well as with mathematical performance (McDougal et al., 2024; 2023; Jirout & Newcombe, 2015).

We introduce **LEGO-Puzzles**, a benchmark designed to systematically evaluate the spatial reasoning capabilities of MLLMS from basic spatial understanding to complex multi-step spatial reasoning. LEGO-Puzzles contains two task sets: The **Elementary** set and the **Planning** set.

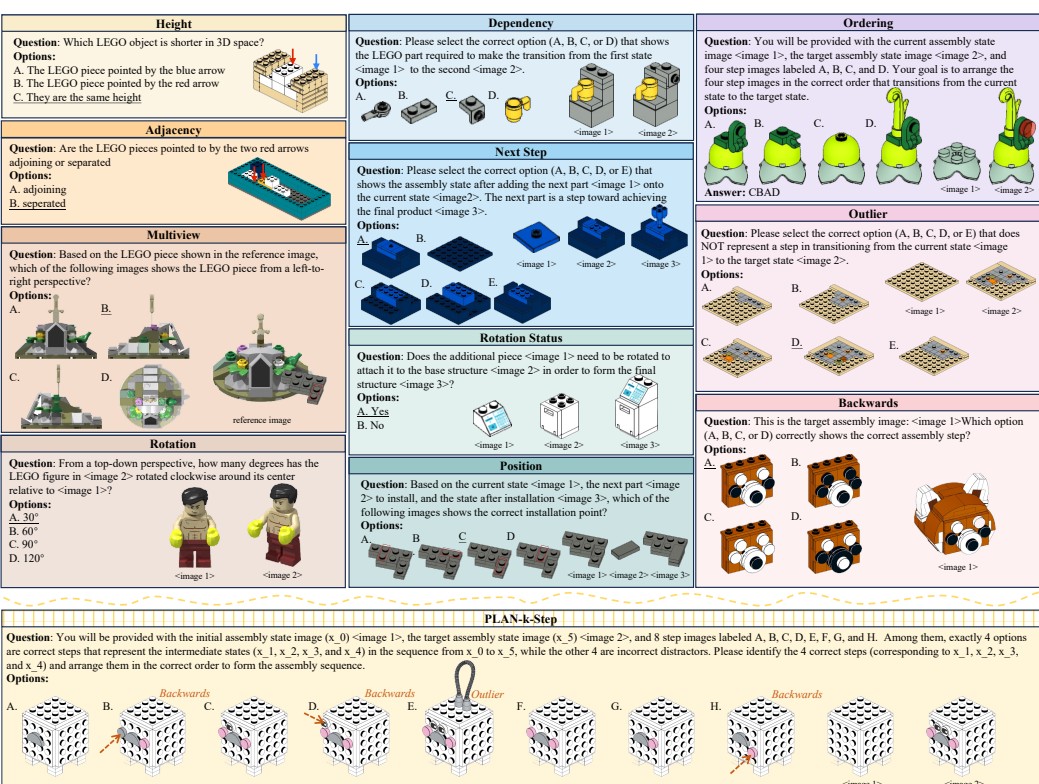

Figure 1: **Task examples of LEGO-Puzzles**. The examples above the wavy line correspond to the *Elementary Set*, with columns (from left to right) representing Spatial Understanding, Single-Step Sequential Reasoning, and Multi-Step Sequential Reasoning. The examples below the wavy line correspond to the *Planning Set*. Note: *The questions are slightly simplified for clarity and brevity.*

The **Elementary** set contains a diverse collection of over 1,100 carefully curated visual question-answering (VQA) pairs spanning 11 distinct tasks, grouped into three levels (Fig. 1). The first level assesses MLLMs' **basic spatial understanding** capabilities, including recognition of height relationships, rotation angles, etc. The second and third levels test **single-step** and **multi-step sequential reasoning** abilities of MLLMs based on LEGO assembly sequences to examine models' sequential reasoning ability. The **Planning** set is especially designed to assess whether a model's capability of multi-step spatial reasoning is strong enough to generate an explicit multi-step plan for assembling a target LEGO structure, with the number of required planning steps varying from 1 to 8. Together, these two sets provide a progressively structured evaluation that spans from basic spatial understanding to long-horizon planning.

Leveraging LEGO-Puzzles, we conduct comprehensive evaluations of 23 state-of-the-art MLLMs, including proprietary models such as GPT-5 and Gemini-2.5-Pro, as well as leading open-source alternatives (Chen et al., 2024b; Wang et al., 2024a; Wu et al., 2024; OpenBMB, 2024).Our experimental results show that even the strongest models struggle with elementary reasoning tasks, falling at least 20% behind human performance. Among open-source models, only a few achieve performance notably above random guessing across different tasks. The planning accuracy also quickly drops to 0% as the number of steps increases. When the number of planning steps exceeds 3, the accuracy of all models drops below 50%. By contrast, human participants solve all tasks perfectly, underscoring the gap between current MLLMs and human spatial reasoning.

Another interesting observation from our experiments is that changing the output format of LEGO-Puzzles tasks from multiple choice to image generation significantly reduces the performance of MLLMs to near zero. In these generation tests, most of the evaluated models fail completely, either disregarding the provided instructions or generating completely irrelevant images.

In summary, LEGO-Puzzles provides a comprehensive evaluation of the spatial understanding and sequential reasoning capabilities of MLLMs. Our main contributions are as follows:

• **Progressive and comprehensive task coverage.** Our benchmark includes a diverse set of tasks spanning basic spatial understanding, single-step reasoning, and multi-step reasoning. This enables systematic evaluation of MLLMs' reasoning capabilities across increasing levels of spatial and sequential complexity.

• **Evaluation of multi-step planning.** Built upon LEGO's step-by-step building process, the Planning set of LEGO-Puzzles is explicitly designed to assess the multi-step spatial reasoning capability of MLLMs with the number of required planning steps varying from 1 to 8.

• **Exploratory evaluation of how VQA reasoning transfers to image generation.** Our exploratory evaluation of spatial reasoning capability of MLLMs on image generation tasks shows that their VQA performance does not necessarily transfer to image generation.

## 2 LEGO-PUZZLES

In this section, we introduce LEGO-Puzzles, a diverse and comprehensive benchmark designed to evaluate the multi-step spatial reasoning capability of MLLMs in detail. Specifically, we first introduce the motivation and definition of the tasks in Sec. 2.1, organized into two parts: the **Elementary set** and the **Planning set**. Then, we introduce our dataset curation process, including data collection, question-answer generation, and quality control, in Sec. 2.2.

### 2.1 TASK DEFINITION

#### 2.1.1 ELEMENTARY SET

The Elementary Set is designed to test fundamental saptial reasoning skills that are crucial for LEGO assembly. To enable a more comprehensive and progressively structured evaluation, we define three levels of tasks. This framework is grounded in insights from cognitive psychology and human developmental stages in acquiring spatial intelligence (Newcombe & Frick, 2010; Bornstein, 1986; Willard, 2022). Using LEGO building as a concrete and intuitive example, we observe that humans typically develop spatial reasoning abilities in stages: from basic *spatial understanding (Level 1)*, to reasoning through *individual assembly steps (Level 2)*, and ultimately to reasoning across *multiple sequential steps (Level 3)*. Based on this developmental trajectory, our benchmark is divided into three levels, as illustrated in Fig. 1.

*Level 1:* **Spatial Understanding.** This level focuses on the ability to *understand the spatial relationships* between each LEGO piece and how the pieces appear and relate when viewed from different perspectives in 3D space: (1) *Height:* Distinguish the relative heights of LEGO objects. (2) *Adjacency:* Determine whether LEGO objects are adjacent or separated. (3) *Rotation:* Calculate the angle of rotation between a LEGO object and its corresponding rotated version. (4) *Multiview:* Predict the current LEGO status from different viewpoints.

*Level 2:* **Single-Step Sequential Reasoning.** Building upon spatial understanding, this level is designed based on single-step actions, mirroring how humans typically progress from spatial

perception to carrying out one assembly step at a time during the building process: (5) *Rotation Status:* Determine whether a LEGO piece requires rotation before installation. In contrast to *Rotation Task* in Level 1, this task focuses on reasoning from an assembly perspective, with finer granularity. (6) *Position:* Identify the correct assembly position to place the next LEGO piece. (7) *Next-Step:* Predict the next LEGO status based on the current status and the new pieces. (8) *Dependency:* Identify which pieces are necessary to transition from the current to the next assembly stage.

***Level 3:* Multi-Step Sequential Reasoning.** The final level is built upon the single-step reasoning skills from *Level 2* and require planning over multiple sequences (involving up to 7 intermediate stages). It assesses the ability to reason across multiple steps in the assembly process: (9) *Backwards:* Given the target assembly stage, determine the correct intermediate step in the LEGO build process. (10) *Ordering:* Determine the correct assembly order of the provided final LEGO images. (11) *Outlier:* Detect the LEGO status that does not belong to the provided assembly sequence.

In conclusion, the Elementary set consists of over $1,100$ visual question-answering (VQA) pairs derived from 407 LEGO building instructions, encompassing 11 tasks across spatial understanding, single-step and multi-step sequential reasoning. Each task contains 100 samples to ensure balanced evaluation across categories.

### 2.1.2 PLANNING SET

To further investigate the long-horizon multi-step sequential reasoning abilities of MLLMs, we construct the Planning Set, which goes beyond elementary spatial reasoning by requiring step-by-step plan. Within this set, we design a scalable and challenging task, **PLAN-k-Step**, which integrates the multi-step tasks *Backwrds*, *Ordering* and *Outlier* from the Elementary set to model the realistic reasoning planning and execution across multiple sequences under noisy conditions.

*PLAN-k-Step* first extends the original *Ordering* setting, which was restricted to a fixed *PLAN-4-Step*, into a scalable sequence length $k$ setting ranging from 1 to 8. Next, we add an additional $k$ erroneous options from the tasks *Backwrds* and *Outlier* on each $k$-length sequence, resulting in a total of $2k$ options. These noisy conditions increase the task's complexity and make the examination of MLLMs' multi-step reasoning ability more realistic and challenging. Specifically, we construct 20 test cases for every sequence length k. Each test case includes the current LEGO object ($x_0$), the target LEGO object ($x_{k+1}$), the correct intermediate LEGO objects ($x_1^c, x_2^c, \ldots, x_k^c$), and the erroneous LEGO objects ($x_1^e, x_2^e, \ldots, x_k^e$), along with the corresponding text instructions. The model is required to select the $k$ correct intermediate steps and order them according to the assembly sequence.

### 2.2 DATASET CURATION

As illustrated in Fig. 2, our pipeline consists of three key steps: data collection, question-answer generation, and quality control. This design ensures the scalability, accuracy, and reliability of our data.

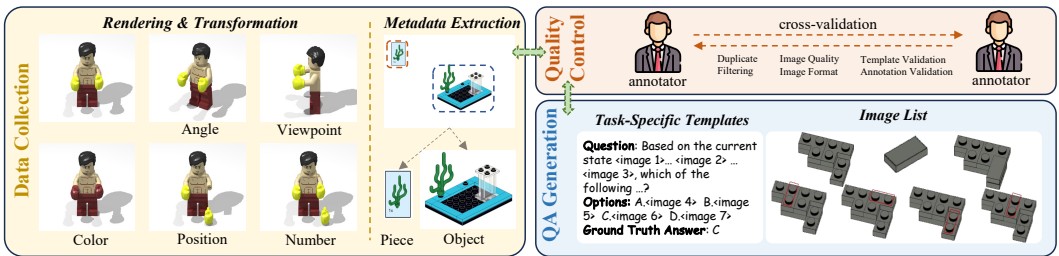

Figure 2: **Data curation pipeline.**

**Data Collection.** Data collection consists of three stages. *(1). LEGO Project Collection.* We collect a diverse set of open-source LEGO source files from the Internet, each containing detailed step-by-step building instructions and part lists. To ensure suitable task complexity, we filter for projects with moderate final size. Extremely large builds exhibit high structural complexity, and the small visual changes introduced by added pieces make it difficult for models to detect step-wise differences. Conversely, overly small builds are filtered out for low spatial complexity and insufficient steps for

multi-step reasoning. We also ensure category diversity, covering animals, furniture, vehicles, and more, to increase task variety. *(2). Rendering and Transformation.* We render each project into PDF format using the public software Studio[1], keeping the camera viewpoint fixed across steps to maintain spatial and temporal consistency. The tool allows flexible editing of the source files, enabling us to modify part attributes such as type, quantity, color, and position as needed for task construction. For example, in the *Rotation* and *Multiview* task, we apply POV-Ray style rendering and adjust lighting to simulate different viewing angles. In the *Backward* task, we introduce deliberate errors in part attributes to generate incorrect assembly states. *(3).Metadata Extraction and Unification.* We employ PDF-Extract[2] to extract structured information from the rendered PDFs, including individual LEGO pieces and assembled objects. All visual assets are processed under a unified naming convention and organized for downstream question-answer generation across all defined task types.

**Question-Answer Generation.** To support scalable and structured data construction, we manually design task-specific templates tailored to each task. Each example includes an instruction defining the model's role, a question referencing input images using tokens like <image x>, and a ground-truth answer. For example, in the *Position* task, the model is provided with the current assembly state, the part to install, the resulting state, and several candidate placements. This standardized template design allows flexibility in the number of input images required by each task, making it suitable for both single-step and multi-step reasoning scenarios. More details are provided in Sec. 8.5.

**Quality Control.** To ensure data quality and reliability, we implement a multi-stage human-in-the-loop review process. *1)Duplicate filtering.* We perform duplication checks by computing similarity scores across rendered images, identifying visually redundant or recolored samples. These are further reviewed manually, and duplicates are removed to reduce redundancy and improve evaluation quality. *2)Image Quality and Format Check.* We manually verify all rendered images to ensure consistency in camera perspective, correctness of part attributes (e.g., color, shape, quantity), and adherence to naming conventions. Any files with errors are either corrected or discarded. *3)Template and annotation validation.* Each question-answer pair is verified by three trained annotators. Reviewers confirm that the images referenced by tokens (<image 1>, <image 2>, ...) appear in the correct order. Samples with unresolved disagreements are either revised or removed.

Given the vast number of high-quality open-source LEGO projects available, our pipeline is inherently scalable and can be extended in an semi-automated manner to support larger and more diverse benchmarks in the future.

## 3 EXPERIMENT

### 3.1 EXPERIMENTAL SETTING

**Benchmark Models.** For the Elementary Set, we extensively evaluate 23 models, covering a diverse range of architectures, sizes, and training processes. For open-source models, we evaluate Qwen2-VL-[7B/72B] (Wang et al., 2024a), Qwen2.5-VL-[7B/72B] (Bai et al., 2025), InternVL2.5-[8B/78B] (Chen et al., 2024a), MiniCPM-V2.6 (Yao et al., 2024), VILA1.5-13B (Lin et al., 2024), Idefics3-8B (Laurençon et al., 2024), DeepSeek-VL2-[Tiny/Small] (Wu et al., 2024), Pixtral-12B (Agrawal et al., 2024), LLaVA-OneVision-7B (Li et al., 2024), and EMU3 (Wang et al., 2024b). For proprietary models, we evaluate GPT-5(20250807) (OpenAI, 2025a), GPT-o3 (OpenAI, 2025c), GPT-4o (20241120), GPT-4o-mini (OpenAI, 2023), Claude-3.5-Sonnet (Anthropic, 2024), Gemini-2.5-Pro (Comanici et al., 2025), Gemini-2.0-Flash, Gemini-1.5-Flash (Team et al., 2023), and Gemini-1.5-Pro. For the Planning Set, we focus on long-horizon reasoning and thus select the four top-performing models from the original *Ordering* task: GPT-5, Gemini-2.5-Pro, Qwen2.5-VL-72B, and InternVL2.5-78B. Moreover, all evaluations are conducted in a zero-shot setting for a fair comparison.

**Baselines.** We provide two baselines for comparison:
- *Random*: the accuracy of random selection, assuming equal probability for all options.
- ↑ *Random*: the *p-value–based critical value*, which indicates the minimum accuracy required to statistically surpass random guessing at a given significance level ($p = 0.05$).

---

[1] https://www.bricklink.com/v3/studio/download.page
[2] https://github.com/opendatalab/PDF-Extract-Kit

Table 1: **Full evaluation results of 23 MLLMs on the Elementary set.** Dark Gray and Light Gray indicates the best performance for each task among all models and open-source models respectively. We highlight the top 3 models by overall performace using Dark Green , Medium Green , and Light Green , respectively.

| Models | Spatial Understanding | | | | Single-Step Reasoning | | | | Multi-Step Reasoning | | | Overall |
|---|---|---|---|---|---|---|---|---|---|---|---|---|
| | Height | Adjacency | Rotation | Multiview | Next-Step | Dependency | Rotation Stat. | Position | Backwards | Ordering | Outlier | |
| *Proprietary* | | | | | | | | | | | | |
| GPT-5 | 62.0 | 69.0 | 72.0 | 67.0 | 83.0 | 92.0 | 55.0 | 59.0 | 73.0 | 85.0 | 75.0 | 72.0 |
| GPT-o3 | 66.0 | 71.0 | 64.0 | 66.0 | 83.0 | 92.0 | 53.0 | 54.0 | 66.0 | 80.0 | 65.0 | 69.1 |
| GPT-4o | 49.0 | 66.0 | 41.0 | 51.0 | 65.0 | 87.0 | 51.0 | 51.0 | 53.0 | 72.0 | 49.0 | 57.7 |
| GPT-4o-mini | 31.0 | 53.0 | 26.0 | 51.0 | 27.0 | 71.0 | 57.0 | 32.0 | 50.0 | 7.0 | 27.0 | 39.3 |
| Claude-3.5-Sonnet | 39.0 | 60.0 | 42.0 | 48.0 | 61.0 | 78.0 | 58.0 | 37.0 | 49.0 | 54.0 | 64.0 | 53.6 |
| Gemini-2.5-Pro | 57.0 | 64.0 | 65.0 | 65.0 | 79.0 | 88.0 | 64.0 | 67.0 | 59.0 | 82.0 | 67.0 | 68.8 |
| Gemini-2.0-Flash | 35.0 | 70.0 | 49.0 | 45.0 | 69.0 | 81.0 | 54.0 | 46.0 | 56.0 | 46.0 | 43.0 | 54.0 |
| Gemini-1.5-Flash | 29.0 | 58.0 | 28.0 | 45.0 | 57.0 | 77.0 | 57.0 | 32.0 | 28.0 | 20.0 | 51.0 | 43.8 |
| Gemini-1.5-Pro | 35.0 | 58.0 | 38.0 | 56.0 | 59.0 | 84.0 | 61.0 | 39.0 | 35.0 | 44.0 | 59.0 | 51.6 |
| *Open-source* | | | | | | | | | | | | |
| Qwen2.5-VL-72B | 30.0 | 61.0 | 27.0 | 27.0 | 55.0 | 72.0 | 58.0 | 47.0 | 60.0 | 33.0 | 43.0 | 46.6 |
| Qwen2.5-VL-7B | 35.0 | 60.0 | 22.0 | 27.0 | 26.0 | 60.0 | 49.0 | 25.0 | 24.0 | 5.0 | 13.0 | 31.5 |
| Qwen2-VL-72B | 40.0 | 62.0 | 37.0 | 51.0 | 57.0 | 79.0 | 49.0 | 43.0 | 34.0 | 26.0 | 31.0 | 46.3 |
| Qwen2-VL-7B | 31.0 | 57.0 | 30.0 | 40.0 | 44.0 | 70.0 | 48.0 | 26.0 | 13.0 | 9.0 | 28.0 | 36.0 |
| InternVL2.5-78B | 41.0 | 62.0 | 32.0 | 47.0 | 60.0 | 79.0 | 58.0 | 32.0 | 40.0 | 15.0 | 37.0 | 45.7 |
| InternVL2.5-8B | 35.0 | 53.0 | 23.0 | 37.0 | 38.0 | 48.0 | 64.0 | 25.0 | 35.0 | 0.0 | 29.0 | 35.2 |
| MiniCPM-V2.6 | 26.0 | 56.0 | 22.0 | 44.0 | 34.0 | 50.0 | 51.0 | 29.0 | 23.0 | 0.0 | 19.0 | 32.2 |
| VILA1.5-13B | 26.0 | 55.0 | 26.0 | 35.0 | 17.0 | 34.0 | 48.0 | 26.0 | 12.0 | 4.0 | 22.0 | 27.7 |
| Idefics3-8B | 29.0 | 51.0 | 23.0 | 23.0 | 18.0 | 20.0 | 47.0 | 30.0 | 24.0 | 4.0 | 24.0 | 26.6 |
| DeepSeek-VL2-Small | 31.0 | 52.0 | 36.0 | 41.0 | 38.0 | 57.0 | 59.0 | 28.0 | 41.0 | 3.0 | 26.0 | 37.5 |
| DeepSeek-VL2-Tiny | 32.0 | 52.0 | 36.0 | 24.0 | 27.0 | 25.0 | 47.0 | 27.0 | 26.0 | 4.0 | 16.0 | 28.7 |
| Pixtral-12B | 31.0 | 68.0 | 24.0 | 24.0 | 21.0 | 38.0 | 53.0 | 21.0 | 24.0 | 3.0 | 37.0 | 31.3 |
| LLaVA-OneVision-7B | 42.0 | 59.0 | 21.0 | 41.0 | 30.0 | 50.0 | 59.0 | 26.0 | 20.0 | 0.0 | 22.0 | 33.6 |
| EMU3 | 31.0 | 52.0 | 24.0 | 25.0 | 17.0 | 25.0 | 47.0 | 25.0 | 24.0 | 0.0 | 20.0 | 26.4 |
| *Baseline* | | | | | | | | | | | | |
| Random Guessing | 33.0 | 50.0 | 25.0 | 25.0 | 20.0 | 25.0 | 50.0 | 25.0 | 25.0 | 4.2 | 20.0 | 27.5 |
| ↑ Random ($p < 0.05$) | 42.0 | 59.0 | 33.0 | 33.0 | 28.0 | 33.0 | 59.0 | 33.0 | 33.0 | 9.0 | 28.0 | 35.5 |

Table 2: **Comparing top-performing MLLMs with human proficiency on LEGO-Puzzles-Lite.** The best results are marked in **bold**. The top 3 overall performances are highlighted in Dark Green , Medium Green , and Light Green , respectively.

| Models | Spatial Understanding | | | | Single-Step Reasoning | | | | Multi-Step Reasoning | | | Overall |
|---|---|---|---|---|---|---|---|---|---|---|---|---|
| | Height | Adjacency | Rotation | Multiview | Next-Step | Dependency | Rotation Stat. | Position | Backwards | Ordering | Outlier | |
| *LEGO-Puzzles-Lite* | | | | | | | | | | | | |
| **Human proficiency** | 70.0 | 95.0 | 95.0 | 100.0 | 90.0 | 100.0 | 100.0 | 95.0 | 95.0 | 95.0 | 95.0 | 93.6 |
| GPT-5 | 55.0 | 70.0 | 80.0 | 85.0 | 75.0 | 95.0 | 55.0 | 60.0 | 70.0 | 85.0 | 60.0 | 71.8 |
| GPT-o3 | 65.0 | 70.0 | 75.0 | 70.0 | 75.0 | 95.0 | 55.0 | 55.0 | 80.0 | 75.0 | 65.0 | 70.9 |
| Gemini-2.5-Pro | 40.0 | 70.0 | 70.0 | 55.0 | 85.0 | 90.0 | 50.0 | 50.0 | 55.0 | 80.0 | 80.0 | 65.9 |
| Qwen2.5-VL-72B | 25.0 | 70.0 | 25.0 | 35.0 | 65.0 | 70.0 | 65.0 | 45.0 | 55.0 | 20.0 | 55.0 | 48.2 |
| InternVL2.5-78B | 40.0 | 55.0 | 30.0 | 45.0 | 60.0 | 85.0 | 55.0 | 30.0 | 25.0 | 20.0 | 50.0 | 45.0 |
| Qwen2-VL-72B | 30.0 | 65.0 | 45.0 | 50.0 | 55.0 | 80.0 | 45.0 | 35.0 | 30.0 | 15.0 | 35.0 | 44.1 |

**Evaluation Metrics.** For all questions in the Elementary set, we adopt *exact match accuracy* (%) as the evaluation metric. For multiple-choice questions, we follow VLMEvalKit (Duan et al., 2024), applying rule-based matching as a first step, and resort to LLM-based choice extraction and matching using ChatGPT (OpenAI, 2023) when heuristic matching fails. For the *Ordering* task, we adopt the same strategy: rule-based extraction of the predicted step sequence from the model's response, followed by LLM-based extraction and matching when necessary. For the Planning set, we use *exact match accuracy* as in the Elementary set, and additionally include two metrics: (1) *set match accuracy*(%), defined as the proportion of predictions in which the predicted steps contain exactly the same elements as the ground truth, regardless of order. (2) *overlap ratio* (%), which measures the fraction of predicted steps that also appear in the ground-truth sequence without considering order.

## 3.2 MAIN RESULTS

### 3.2.1 THE ELEMENTARY SET

We include evaluation results for the Elementary Set in Tab. 1 and Tab. 2. We summarize key findings as below.

**Clear Gap Exists Between Proprietary and Open-source models.** There is a significant gap between open-source and proprietary MLLMs in both spatial understanding and sequential reasoning abilities as shown in Tab. 1. Most open-source MLLMs perform only marginally better than *Random*, while leading proprietary models, such as Gemini-2.5-Pro and GPT-5, exhibit strong spatial reasoning capabilities, achieving overall accuracies of 68.8% and 72.0%, respectively.

**Human Outperforms the Best MLLMs by Over 20%.** To study the performance gap between human and MLLMs on the Elementary set of LEGO-Puzzles, we randomly select 20 questions from each task to create LEGO-Puzzles-Lite, resulting in a total of 220 QA pairs, and use this subset for investigation. We invited 30 human experts to solve these questions, ensuring a diverse and representative evaluation of human-level reasoning. Our findings indicate that human experts consistently achieve significantly higher overall performance (93.6%), as shown in Tab. 2. In contrast, current MLLMs fall short, with even the most advanced models, Gemini-2.5-pro and GPT-5, trailing over 20% behind human performance across all tasks. This persistent gap highlights the need for comprehensive and substantial improvements in the ability of MLLMs.

**MLLMs Struggle to 3D Spatial Reasoning.** Our results reveal that MLLMs struggle significantly with tasks requiring true 3D spatial reasoning. In the *Height* task, we deliberately construct cases where interpreting the image from a 2D viewpoint leads to a different answer than interpreting it from a true 3D perspective. As shown in Tab. 1, most models (11/23) perform worse than *Random*. These results suggest that MLLMs tend to defaultly answer questions based on a 2D projection rather than a true 3D perspective. This observation highlights their reliance on 2D spatial priors during inference, underscoring the need for further research on equipping models with robust 3D spatial reasoning capabilities. Similarly, in the *Rotation* task, which requires identifying the rotation angle of an object, 7 out of 23 models fall below *Random*, with most failing to reach even 40% accuracy. This further indicates that current MLLMs struggle to perceive and distinguish object orientation changes reliably.

**MLLMs Perform Poor in Multi-Step Reasoning.** LEGO-Puzzles also reveals substantial limitations in MLLMs' sequential reasoning capabilities, particularly when multiple reasoning steps are required. While performance on single-step tasks such as *Dependency* and *Next-Step* is relatively stronger, models struggle considerably with multi-step tasks. As shown in Tab. 1, almost half of the models score below ↑ *Random* in the *Ordering* task, with some models (e.g., InternVL2.5-8B, LLaVA-OneVision-7B, EMU3) failing completely. Similar trends are observed in the *Backwards* task, where 9 out of 14 open-source models perform below ↑ *Random*. To further explore the upper bound of MLLMs in multi-step reasoning, we conduct the more challenging and length-controllable task *PLAN-k-Step*, as described in Sec. 2.1.2.

In conclusion, the Elementary set of LEGO-Puzzles highlights both the spatial understanding and sequential reasoning abilities of MLLMs. The overall results suggest significant room for improvement, particularly in domains involving relative relationships, rotation perception, and long-range sequential reasoning.

### 3.2.2 THE PLANNING SET

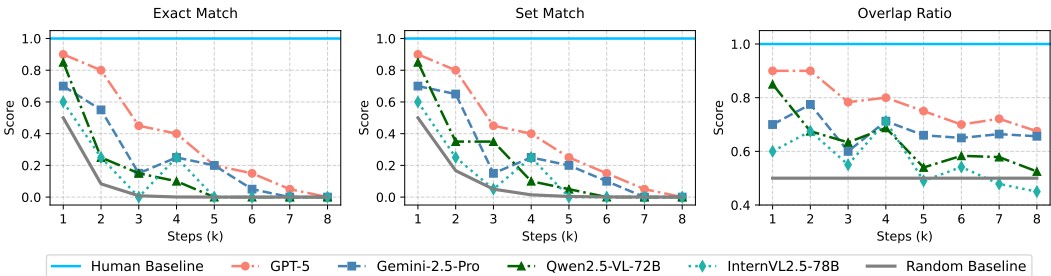

Figure 3: **Evaluation on *PLAN-k-Step*** of different MLLMs across varying numbers of steps.

**Performance Degradation when $k$ Increases.** As shown in Fig. 3, all models exhibit a clear decline in *exact match accuracy* as the number of planning steps $k$ increases. GPT-5 drops from 90% at $k = 1$ to 0% at $k = 8$, while both Qwen2.5-VL-72B and InternVL2.5-78B reach 0% once $k > 4$. For $k > 3$, the accuracy of all models drops below 50%.

**Model Struggles in Selecting the Correct Set of Steps.** Comparing the accuracy of *exact match* and *set match*, GPT-5 exhibits almost identical curves. This indicates that errors emerge already at the set-selection stage: the model fails to identify the correct elements and consequently selects an incorrect set of steps, let alone ordering them correctly.

**Clear Gap Between Open-Source and Proprietary Models.** For *overlap ratio*, even when *exact match accuracy* falls to 0% at $k = 7/8$, the proprietary models still achieve $\sim 65\%$, indicating that they often identify a substantial portion of the correct steps. Conversely, the open-source models degrade to near random-guessing levels as $k > 4$.

**MLLMs Perform Far Below Human Level.** To establish a human baseline, we invited five expert participants to solve the same set of tasks. Humans achieved a perfect score of 100% across all values of $k$, showing no decline as the number of steps increased. On average, they completed all tasks in about 85 minutes. This demonstrates that, unlike MLLMs, humans can reliably handle long-horizon multi-step reasoning tasks. In contrast, current MLLMs struggle to track and integrate spatial transformations across multiple steps, where accumulated errors lead to inconsistent predictions in longer reasoning chains.

## 4 IMAGE GENERATION EVALUATION

In Sec. 3.2, we use LEGO-Puzzles to evaluate MLLMs in a multiple-choice setting, where models select from predefined options. In this section, to further investigate whether MLLMs can transfer their spatial reasoning abilities to image generation, we change the output format of LEGO-Puzzles tasks from multiple choice to image generation, where models are required to directly produce visual outputs instead of selecting from given options. We select five tasks—*Rotation\**, *Multiview\**, *Position\**, *Dependency\**, and *Next-Step\** from LEGO-Puzzles—resulting in a total of 100 questions.

We evaluate the open-source models Emu2 (Sun et al., 2023), GILL (Koh et al., 2023), and Anole (Chern et al., 2024), as well as the proprietary models GPT-4o\*, GPT-4o and Gemini-2.0-Flash, all of which support long-range sequence input and image output. For evaluation, traditional metrics such as FID (Heusel et al., 2017), CLIPScore (Hessel et al., 2021; Gao et al., 2024), and X-IQE (Chen et al., 2023) mainly assess image fidelity or cross-modal alignment, often relying on pre-trained model priors or fixed scoring heuristics. They struggle to capture the fine-grained spatial accuracy required in LEGO assembly tasks, where even small errors—such as misaligned parts or incorrect orientations—can invalidate the result. Furthermore, many recent multimodal evaluation metrics depend on GPT-based models (Liu et al., 2024a), introducing uncontrollable bias into the evaluation process. Therefore, we enlist 5 human experts to assess model performance across two dimensions: appearance similarity and instruction following. Each aspect is rated on a scale from 0 to 3. Detailed scoring guidelines are provided in Sec. 8.4.2.

Table 3: **Evaluation on *generation*.** We conduct human-based evaluation to assess the "Appearance" (App) and "Instruction Following" (IF) scores of Gemini-2.0-Flash, GPT-4o, Emu2, GILL, and Anole, using a scoring scale from 0 to 3 for both dimensions.

| Task \MLLM | GPT-4o | | Gemini-2.0-Flash | | GPT-4o* | | Emu2 | | GILL | | Anole | |
|---|---|---|---|---|---|---|---|---|---|---|---|---|
| | App | IF | App | IF | App | IF | App | IF | App | IF | App | IF |
| Rotation* | 2.35 | 1.75 | 2.05 | 1.45 | 0.65 | 0.50 | 1.70 | 0.00 | 0.00 | 0.00 | 0.10 | 0.00 |
| Multiview* | 2.30 | 2.00 | 2.40 | 1.65 | 1.95 | 0.40 | 1.65 | 0.25 | 0.00 | 0.00 | 0.05 | 0.00 |
| Position* | 2.30 | 1.65 | 2.85 | 1.30 | 2.95 | 1.00 | 0.50 | 0.00 | 0.00 | 0.00 | 0.00 | 0.00 |
| Dependency* | 2.05 | 2.00 | 1.70 | 0.95 | 0.35 | 0.05 | 0.55 | 0.00 | 0.00 | 0.00 | 0.00 | 0.00 |
| Next-Step* | 2.25 | 1.45 | 1.75 | 0.05 | 0.30 | 0.00 | 0.05 | 0.00 | 0.00 | 0.00 | 0.00 | 0.00 |
| Overall | 2.25 | 1.77 | 2.15 | 1.08 | 1.24 | 0.39 | 0.89 | 0.05 | 0.00 | 0.00 | 0.03 | 0.00 |

Tab. 3 presents the human evaluation results across five generation tasks. Overall, proprietary models outperform open-source ones in both appearance consistency (App) and instruction adherence (IF). Among them, GPT-4o achieves the highest overall scores (App: 2.25, IF: 1.77), followed by Gemini-2.0-Flash (App: 2.15, IF: 1.08). However, both models show clear room for improvement in instruction following, scoring only 1.77 and 1.08 out of 3, respectively. For GPT-4o\*, the results suggest that the model may not directly edit the input image, but instead reinterpret the scene

---

\*refers to the version of GPT-4o released prior to March 6, 2025.

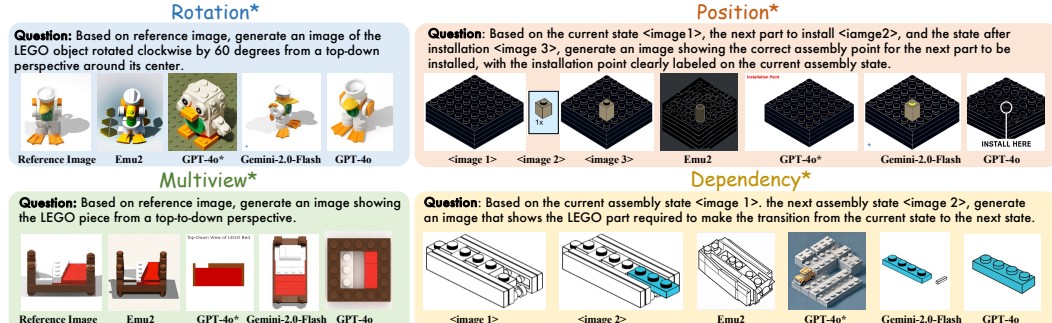

Figure 4: **Qualitative visual results for image generation tasks.** Note: The questions above are slightly simplified for clarity and brevity.

semantically and regenerate it based on textual understanding. This leads to lower appearance consistency (App: 1.24), reflecting a conceptual reconstruction process rather than precise visual editing. Among open-source models, Emu2 shows some ability to preserve visual appearance (App: 0.89) but fails almost entirely in instruction following (IF: 0.05), treating the task more as image replication than reasoning-based generation. GILL and Anole perform the worst, with near-zero scores across all tasks and frequently irrelevant outputs. The qualitative results are shown in Fig. 4. More cases are provided in Sec. 8.6.

Overall, these results show that current models—especially open-source ones—struggle significantly with instruction-grounded image generation, highlighting the challenges of spatially grounded visual synthesis.

## 5  DISCUSSION

Table 4: Pearson correlation co-efficients (PCC) and p-values for *height* and *adjacency* tasks.

| Task | PCC | P-value |
|------|-----|---------|
| Height | 0.93 | 0.00723 |
| Adjacency | 0.98 | 0.00046 |

While LEGO-Puzzles is built on rendered data, it aims to evaluate fundamental spatial reasoning capabilities that are also essential in real-world scenarios. To assess its generalizability beyond synthetic environments, we compare model performance on LEGO-Puzzles with 3DSRBench (Ma et al., 2024), a benchmark based on natural images. Both datasets contain conceptually similar tasks—specifically, the *Height* task in LEGO-Puzzles aligns with *Height* in 3DSRBench, and *Adjacency* in LEGO-Puzzles corresponds to the *Location* task in 3DSRBench.

We evaluate all proprietary models tested in LEGO-Puzzles on the corresponding tasks in 3DSRBench and compute the Pearson correlation coefficient (Cohen et al., 2009) to measure consistency in performance across the two datasets. As shown in Tab. 4, the results reveal strong positive correlations: 0.93 for *Height* and 0.98 for *Adjacency*, both statistically significant (p < 0.01).

These findings suggest that LEGO-Puzzles not only offers high scalability and precise control in synthetic settings but also captures spatial reasoning patterns that generalize well to natural images. This validates its utility as a proxy for evaluating real-world spatial understanding.

## 6  RELATED WORK

**General Multi-Modal Evaluation Benchmarks.** Recent years have seen significant advancements in MLLMs, accompanied by a surge in benchmark datasets evaluating their visual understanding. MME (Fu et al., 2023) provides a systematic evaluation of 14 image-centric tasks, revealing persistent challenges such as object hallucination and spatial reasoning failures. MMBench (Liu et al., 2024b) introduces a bilingual multiple-choice format for fine-grained multimodal assessment. Moving beyond static images, SEED-Bench (Li et al., 2023a) evaluates generative comprehension spanning both image and video reasoning. For expert-level reasoning, MMMU (Yue et al., 2024) presents a discipline-specific benchmark across 183 subtopics, revealing substantial knowledge gaps even in leading MLLMs. Overall, these benchmarks reveal that while MLLMs have made progress, they still struggle with spatial understanding and long-horizon reasoning, presenting clear directions for future research.

**Visual Spatial Reasoning Evaluation Benchmarks.** Multimodal large language models (MLLMs) have made notable progress on vision-and-language tasks, yet they still struggle with 3D spatial reasoning. Beyond general benchmarks, several more specific datasets have been proposed to evaluate the spatial reasoning abilities of MLLMs. CLEVR (Johnson et al., 2017) evaluates diverse visual reasoning skills with minimal biases and fine-grained annotations. More recently, 3DSRBench (Ma et al., 2024) designs 12 tasks assessing spatial understanding from perspectives such as relative height, location, and orientation. SpatialViz-Bench (Wang et al., 2025) provides a cognitively grounded benchmark of 12 tasks targeting four spatial skills: rotation, folding, penetration, and animation. Despite their significance, most of these efforts primarily focus on *single-image* spatial reasoning. Several benchmarks incorporate sequential aspects of spatial reasoning. VSI-Bench (Yang et al., 2024) introduces a video-based benchmark evaluate MLLMs' visual-spatial intelligence from sequential observations. ActPlan-1K (Su et al., 2024) focuses on procedural planning for household activities given images and task descriptions. PhyBlock (Ma et al., 2025) designs a progressive benchmark to assess VLMs on physical understanding and planning. Despite their contributions, these benchmarks either involve overly simplified or highly abstract environments, or lack rigorous evaluation of explicit multi-step planning. Extending beyond prior benchmarks, our work establishes a testbed that directly evaluates MLLMs' fine-grained multi-step spatial reasoning abilities, offering a more comprehensive assessment of long-horizon spatial reasoning.

## 7 CONCLUSION

We introduce LEGO-Puzzles, a novel benchmark specifically designed to evaluate spatial understanding, as well as single-step and multi-step sequential reasoning in MLLMs. Inspired by human cognitive patterns in LEGO construction, we create two task sets. The Elementary set includes over 1,100 carefully curated visual question-answering (VQA) samples across 11 distinct tasks, providing diverse scenarios to assess multimodal visual reasoning. The Planning set directly requires the model to generate a step-by-step plan for assembling a target LEGO structure. We conduct comprehensive experiments with 23 advanced MLLMs, revealing substantial performance gaps compared to humans, particularly in long-horizon planning and the generation of spatially coherent visual outputs. These findings underscore the urgent need to enhance the spatial understanding and sequential reasoning capabilities of multimodal AI.

## ETHICS STATEMENT

In developing *LEGO-Puzzles*, a novel benchmark specifically designed to evaluate spatial understanding, as well as single-step and multi-step sequential reasoning in MLLMs, we are dedicated to upholding ethical standards and promoting responsible AI use. In the data curation pipeline, we strictly follow the state permission for use of LEGO assets on their respective websites. Our code and benchmark will be publicly released to encourage responsible use in MLLMs evaluation, while discouraging unethical practices, including misinformation and harassment. We also advocate for continued research on safeguards and detection mechanisms to prevent misuse and ensure adherence to ethical guidelines and legal frameworks.

## REPRODUCIBILITY STATEMENT

To facilitate replication, we provide additional evaluaion details in Appendix 8.4, including the evaluation prompts we use and scoring guidelines. The data curation pipeline of LEGO-Puzzles is described in Section 2.2. All code, prompts and datasets will be released publicly.

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

# 8 APPENDIX

## 8.1 THE USE OF LARGE LANGUAGE MODELS (LLMS)

We use LLMs to assist in the writing and polishing of this manuscript. Specifically, we employed an LLM to refine language, improve readability, and enhance clarity in several sections. The model was used for tasks such as sentence rephrasing and improving the overall flow of text.

It is important to emphasize that the LLM was not involved in the ideation, research methodology, data collection, analysis, or experimental design. All research concepts, ideas, and results were entirely developed by the authors. The contributions of the LLM were restricted to improving the linguistic quality of the manuscript, with no influence on the scientific content or validity of the work.

The authors take full responsibility for the content of the paper, including any portions generated or refined with the help of the LLM. We further confirm that the use of the LLM complies with ethical guidelines and does not constitute plagiarism or scientific misconduct.

## 8.2 ERROR CASES IN *Height* TASK

The *Height* task in the Elementary set contains intentional 2D–3D ambiguities that can lead to occasional human mistakes. In these cases, humans may rely on quick visual intuition rather than careful spatial reasoning—especially when the blocks are not adjacent and the height must be inferred through spatial imagination and reasoning rather than direct comparison. When objects are farther apart or partially occluded, people may overlook subtle depth differences and misjudge the true height. We also found that these mistakes largely come from not inspecting the images carefully; when participants were explicitly reminded to pay closer attention, they achieved 100]% accuracy. Several such cases are illustrated in Fig. 5.

# Height

**Question:** Which LEGO object is shorter in 3D space?

**Options:**
A. The LEGO piece pointed by the blue arrow
B. The LEGO piece pointed by the red arrow
*C. They are the same height.*

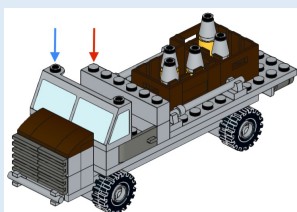

**Question:** Which LEGO object is shorter in 3D space?

**Options:**
A. The LEGO tree marked with a red rectangle.
B. The LEGO tree marked with a blue rectangle.
*C. They are the same height.*

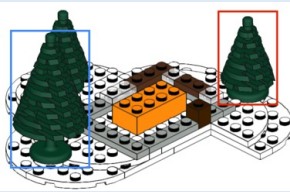

Figure 5: **Error cases in the Height task**. Correct answers are underlined, and *human-selected answers are shown in italics. Note: Questions are slightly simplified for clarity and brevity.*

## 8.3 PROBLEM STATISTICS IN LEGO-PUZZLES

As illustrated in Fig. 6, the Elementary Set contains three levels.

*Level 1:* **Spatial Understanding.** (1) *Height* (2) *Adjacency* (3) *Rotation* (4) *Multiview*

*Level 2:* **Single-Step Sequential Reasoning.** (5) *Rotation Status* (6) *Position* (7) *Next-Step* (8) *Dependency*

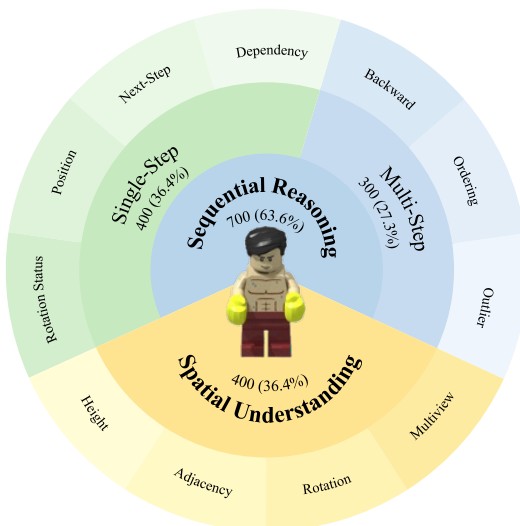

Figure 6: **Problem statistics in the Elementary Set of LEGO-Puzzles**

*Level 3:* **Multi-Step Sequential Reasoning.** (9) *Backwards* (10) *Ordering* (11) *Outlier*

The Elementary set consists of over $1,100$ visual question-answering (VQA) pairs derived from 407 LEGO building instructions, encompassing 11 tasks across spatial understanding, single-step and multi-step sequential reasoning. Each task contains 100 samples to ensure balanced evaluation across categories.

## 8.4 EVALUATION

In this section, we provide the full set of prompt templates used in our evaluation of spatial and sequential reasoning capabilities in MLLMs. All prompts match the requirements of each task type and ensure consistent formatting, minimal ambiguity, and controlled instruction scope. Below, we organize the prompts by task group.

### 8.4.1 PROMPTS USED FOR LEGO-PUZZLES EVALUATION

This section lists the prompt templates used in the LEGO-Puzzles benchmark. Each template corresponds to one of the eleven core tasks defined Sec. 2.1, including *Spatial Understanding*, *Single-step Sequential Reasoning*, and *Multi-step Sequential Reasoning*. For each task, the prompt defines the model's role, describes the input structure (including image tokens), and specifies the required response format.

---

**Prompt for Task *Height***

```
You are a specialized LEGO 3D height analyzer.  Your primary task
is to compare the heights of LEGO objects based on their positions
in 3D space.  You will be provided with an image containing 3D LEGO
objects.  Your answers should be based solely on the provided LEGO 3D
data, without any additional assumptions.  Keep your responses clear,
direct, and focused on the question.  Please respond with only the
letter corresponding to your choice (A, B, or C).\n\n LEGO object is
shorter in 3D space?\n\nHere is the 3D LEGO scene:<image 1> Options:
A.The LEGO tree marked with a red rectangle.  B.The LEGO tree marked
with a blue rectangle.  C.They are the same height.  Please select the
correct answer from the options above.\n
```

---

**Prompt for Task *Adjacency***

You are a specialized LEGO 3D adjacency analyzer. You will be provided with an image containing 3D LEGO objects. Your goal is to determine whether two LEGO objects are directly touching (adjoining) or not (separated) based on their positions in 3D space. Your answers should be based solely on the provided LEGO 3D data, without any additional assumptions. Keep your responses clear, direct, and focused on the question. Please respond with only the letter corresponding to your choice (A or B).\n\nAre the LEGO trees pointed to by the two red arrows adjoining or separated?\n\nHere is the 3D LEGO scene:<image 1>Options: A. Adjoining. B.Seperated. Please select the correct answer from the options above.\n

**Prompt for Task *Multiview***

You are a specialized LEGO 3D multi-view analyzer. Your primary task is to identify the correct perspective of LEGO object images in a 3D scene. You will be provided with one reference image (x_0) and four optional images showing the LEGO object from different viewpoints. Your goal is to determine which option corresponds to one of the following perspectives: top-down, left-to-right, right-to-left, or front-to-back. Your answers should be based solely on the provided LEGO 3D data, without any additional assumptions. Keep your responses clear, direct, and focused on the question. Please respond with only the letter corresponding to your choice (A, B, C, or D).\n\nBased on the LEGO object shown in the reference image (x_0), which of the following images shows the LEGO object from a front-to-back perspective?\n\nReference image (x_0): <image 1>Options: A.<image 2>B.<image 3>C.<image 4>D.<image 5>Please select the correct answer from the options above.\n

**Prompt for Task *Rotation***

You are a specialized LEGO 3D rotation analyzer. You will be provided with two images: Image 1 and Image 2. Image 2 shows the same LEGO object as Image 1, but rotated clockwise around its center from a top-down perspective (looking down on the LEGO object from above) by one of the following angles: 30°, 60°, 90°, or 120°. Your task is to determine how many degrees the LEGO object in Image 2 has rotated clockwise relative to Image 1. Your answers should be based solely on the provided LEGO 3D data, without any additional assumptions. Keep your responses clear, direct, and focused on the question. Please respond with only the letter corresponding to your choice (A, B, C, or D).\n\nHow many degrees has the LEGO object in Image 2 rotated clockwise around its center relative to Image 1?\n\nImage 1 (original): <image 1>Image 2 (rotated): <image 2>Options:A.30° B.60° C.90° D.120° Please select the correct answer from the options above.\n

**Prompt for Task *Dependency***

You are a specialized LEGO 3D assembly analyzer. Your primary task is to identify the correct LEGO piece needed to transition from one assembly state to another. You will be provided with two consecutive assembly state images: the current assembly state ($x_1$) and the next assembly state ($x_2$), followed by four LEGO piece images. Your goal is to select the correct option that shows the LEGO piece required to make the transition from the first state to the second. Your answers should be based solely on the provided sequence of images, focusing on the logical relationships between the sequence of images and the 3D spatial relationships between the LEGO pieces. Avoid making any additional assumptions beyond the information conveyed in the sequence. Keep your responses clear, direct, and focused on the question. Please respond with only the letter corresponding to your choice (A, B, C, or D).\n\nThis is the current assembly state ($x_1$): <image 1>This is the next assembly state ($x_2$): <image 2>\n\nPlease choose which option correctly shows the LEGO piece needed for the transition: Options: A.<image 3>B.<image 4>C.<image 5>D.<image 6> Please select the correct answer from the options above.\n

**Prompt for Task *Next-Step***

You are a LEGO 3D next step prediction analyzer. Your primary task is to identify the correct next state of a LEGO object assembly. You will be provided with the current assembly state ($x_1$), the next LEGO piece to be added ($x_2$), and the target object image ($x_3$), which represents the final completed object. Your goal is to select the correct option that shows the assembly state after adding the next LEGO piece. The next LEGO piece ($x_2$) is a step toward achieving the final object ($x_3$), and your task is to determine how adding this LEGO piece could potentially move the assembly toward the final object. Your answers should be based solely on the provided sequence of images, focusing on the logical relationships between the sequence of images and the 3D spatial relationships between the parts. Avoid making any additional assumptions beyond the information conveyed in the sequence. Keep your responses clear, direct, and focused on the question. Please respond with only the letter corresponding to your choice (A, B, C, D, or E).\n\nThis is the current assembly state ($x_1$):<image 1>This is the next LEGO piece to be used ($x_2$):<image 2>This is the target object ($x_3$):<image 3>Please choose which option correctly shows the state of the object after adding the specified LEGO piece: Options: A.<image 4>B.<image 5>C.<image 6>D.<image 7>E.<image 8> Please select the correct answer from the options above.\n

**Prompt for Task *Rotation Status***

You are a specialized LEGO 3D rotation and assembly analyzer. You will be provided with three images: the base structure ($x_0$), the final structure ($x_1$), and the additional piece ($x_2$). Your task is to analyze the images and decide whether rotating the additional piece ($x_2$) is necessary to attach it to the base structure ($x_0$) and form the final structure ($x_1$). Your answers should be based solely on the provided LEGO 3D data, without any additional assumptions. Keep your responses clear, direct, and focused on the question. Please respond with only the letter corresponding to your choice (A: 'Yes' or B: 'No').\n\nDoes the additional piece ($x_2$) need to be rotated to attach it to the base structure ($x_0$) in order to form the final structure ($x_1$)? Base structure ($x_0$): <image 1>Final structure ($x_1$): <image 2>Additional piece ($x_2$): <image 3> Options: A.Yes. B.No. Please select the correct answer from the options above.\n

**Prompt for Task *Position***

You are a LEGO 3D assembly position analyzer. Your primary task is to determine the correct assembly point of a given LEGO piece based on the current state and the next state. You will be provided with images representing the current state (x_0), the LEGO piece to install (x_1), the state after installation (x_2), and installation options. Your goal is to analyze the given images and determine which of the four options shows the correct assembly point for the next LEGO piece. Your answers should be based solely on the provided LEGO 3D data, without any additional assumptions. Keep your responses clear, direct, and focused on the question. Please respond with only the letter corresponding to your choice (A, B, C, or D).\n\nBased on the current state (x_0), the next LEGO piece (x_1) to install, and the state after installation (x_2), which of the following images shows the correct installation point?Current state (x_0): <image 1>Part to install (x_1): <image 2>State after installation (x_2): <image 3> Options: A.<image 4>B.<image 5>C.<image 6>D.<image 7>Please select the correct answer from the options above.\n

**Prompt for Task *Ordering***

You are a specialized LEGO 3D step ordering analyzer. Your primary task is to order the correct sequence of LEGO assembly steps to build a target model. You will be provided with the current assembly state image (x_1), the target assembly state image (x_2), and four step images. Your goal is to arrange the four step images in the correct order (e.g., 'BDAC') that transitions from the current state to the target state. Your answers should be based solely on the provided sequence of images, focusing on the logical relationships between the sequence of images and the 3D spatial relationships between the parts. Avoid making any additional assumptions beyond the information conveyed in the sequence. Keep your responses clear, direct, and focused on the question. Please respond with only the sequence of letters corresponding to your choice (e.g., 'BDAC'). \n\nThis is the current assembly state (x_1): <image 1>\n\nThis is the target assembly state (x_2): <image 2>\n \n Please arrange the following options (A, B, C, and D) in the correct order to transition from the current state to the target state: A.<image 3>B.<image 4>C.<image 5>D.<image 6> Please select the correct answer from the options above.\n

**Prompt for Task *Outlier***

You are a specialized LEGO 3D step sequence analyzer. You will be provided with: the current assembly state image (x_1) and the target assembly state image (x_2), followed by five step images. Four of these steps correctly fit in the sequence needed to transition from x_1 to x_2, while one does NOT belong in this sequence. Your task is to identify the one incorrect step. Your answers should be based solely on the provided sequence of images, focusing on the logical relationships between the sequence of images and the 3D spatial relationships between the parts. Avoid making any additional assumptions beyond the information conveyed in the sequence. Keep your responses clear, direct, and focused on the question. Please respond with only the letter corresponding to your choice (A, B, C, D, or E).\n\nThis is the current assembly state (x_1): <image 1>This is the target assembly state (x_2): <image 2>\n\n Please choose which option does NOT belong in this sequence:Options: A.<image 3>B.<image 4>C.<image 5>D.<image 6>E.<image 7> Please select the correct answer from the options above.\n

---

**Prompt for Task *Backwards***

```
You are a specialized LEGO 3D reverse engineering analyzer.  You will
be provided with the final target assembly image and four step images.
Your goal is to select the correct option that represents the step to
assemble the target model.  The incorrect options may be rejected for
reasons such as:  incorrect LEGO piece colors, incorrect LEGO piece
positions, incorrect LEGO piece orientations, incorrect number of LEGO
pieces, etc.  Your answers should be based solely on the provided
sequence of images, focusing on the logical relationships between
the sequence of images and the 3D spatial relationships between the
LEGO pieces.  Avoid making any additional assumptions beyond the
information conveyed in the sequence.  Keep your responses clear,
direct, and focused on the question.  Please respond with only the
letter corresponding to your choice (A, B, C, or D).\n\n This is
the target assembly image:  <image 1>Which option correctly shows
the correct assembly step?Options:  A.<image 2>B.<image 3>C.<image
4>D.<image 5>Please select the correct answer from the options
above.\n
```

---

### 8.4.2 HUMAN EVALUATION IN GENERATION

To evaluate the quality of model-generated images, we conduct human annotation using a two-dimensional rubric: *Appearance Similarity (App)* and *Instruction Following (IF)*. Each aspect is scored on a scale from 0 to 3, where higher scores reflect closer alignment with the expected visual outcome. Five expert annotators independently score each sample, and final scores are averaged across annotators. Below, we describe the evaluation criteria in detail.

**Appearance Similarity (0–3 points).** This score measures how visually similar the generated image is to the expected ground-truth output, focusing on global and local visual elements such as object shape, part layout, color, and scene background. It does not consider whether the model followed instructions—only how much the output visually resembles the intended result.

- **3 – Nearly identical:** All major and minor details match; only imperceptible or stylistic differences exist.
- **2 – Mostly similar:** Most key features are correctly rendered, with only minor visual discrepancies (e.g., slightly incorrect positioning or color of one part).
- **1 – Partially similar:** Some features are accurate, but major visual elements are incorrect or missing.
- **0 – Dissimilar:** The overall shape, structure, or color is incorrect; the image fails to reflect the expected design in any meaningful way.

**Instruction Following (0–3 points).** This score evaluates how well the generated image complies with the intended transformation, such as adding a specific piece. This criterion assesses logical consistency with the prompt rather than visual consistency. Since instructions differ by task, we define task-specific IF criteria as follows:

**Rotation\***

- **3** – The object is rotated in the correct direction and by the correct angle.
- **2** – The object is rotated in the correct direction but deviates slightly from the target angle.
- **1** – The object is rotated in the wrong direction, or the rotation magnitude is clearly incorrect.
- **0** – No rotation applied or output is irrelevant.

**Multiview\***

- **3** – Viewpoint precisely matches the target perspective.
- **2** – Viewpoint changes in the correct direction but with slight deviation from the target view.
- **1** – Viewpoint change is in the wrong direction (e.g., left vs. right) or clearly inaccurate.

- **0** – Viewpoint is unchanged or output is irrelevant to the prompt.

**Dependency\***

- **3** – Correct part is generated with accurate shape and color.
- **2** – Mostly correct part, with small errors in color or minor structural additions.
- **1** – Part generated is noticeably different in shape or color from ground truth.
- **0** – Generated part is irrelevant or instruction is not followed.

**Position\***

- **3** – Installation point is precisely identified.
- **2** – Predicted location is close to the correct one but slightly offset.
- **1** – Chosen location is far from the ground-truth installation point.
- **0** – No attempt to identify a placement or irrelevant output.

**Next-Step\***

- **3** – Output correctly reflects the structure after the new part is added.
- **2** – Output reflects the instruction generally but contains moderate deviations.
- **1** – Output includes only superficial or incorrect modifications.
- **0** – Output ignores instruction or is unrelated to the task.

To ensure scoring consistency, all annotators were trained with a set of labeled examples before evaluation. Disagreements were resolved via discussion or removed from the analysis if consensus could not be reached.

## 8.5 DATA CURATION

**Question–Answer Generation.** To support scalable and structured benchmark construction, we develop a fully template-based QA generation pipeline tailored to all tasks in LEGO-Puzzles.

*Question Template.* For each task, we manually design a task-specific question format that defines the model's role, introduces the input images through explicit token <image x>, and specifies the expected output form. This ensures that every QA pair adheres to a unified structure while preserving the distinct requirements of each task. Each template consists of:

- **Task-specific instruction.** Defines the model's expected expertise (e.g., "You are a specialized LEGO 3D rotation analyzer."), ensuring consistent reasoning behavior across samples within the same task.
- **Image-referencing token.** All visual inputs are introduced through explicit tokens such as <image x>, providing a standardized way to embed image information across tasks.
- **Task-specific question format.** Specifies the exact objective for the task.

Appendix 8.4.1 lists the full set of Question templates used in LEGO-Puzzles.

*Ground-Truth Answer.* Ground-truth answers are generated automatically from the structured assembly data (for sequential reasoning tasks) or from human annotation (for spatial understanding tasks). For Single-Step and Multi-Step Sequential Reasoning tasks, every LEGO project is constructed step-by-step with precise metadata (e.g., part ID, assembly-step index). Thus, the ground-truth answer for each question can be automatically extracted. For example, in the *Next-Step task*, given the current assembly-step index $k$ and the final target assembly-step index $m$, the ground-truth next state is simply the image corresponding to assembly-step index $k + 1$. This provides a clear and deterministic answer. For Spatial Understanding tasks, each task involves geometric or perceptual judgments that cannot be derived solely from metadata. Their ground-truth answers are therefore obtained through human annotation to ensure correctness and consistency.

## 8.6 IMAGE GENERATION CASES

To better illustrate model behavior in spatial reasoning through image generation, we provide qualitative examples from three representative tasks: *Rotation\**, *Multiview\**, and *Dependency\**. These cases highlight common success and failure modes across models, revealing how different MLLMs handle viewpoint changes, object rotations, and part-level reasoning. In each case, the task setup and the generated responses are visualized, with questions simplified for clarity. Examples are selected to reflect typical model behavior observed during evaluation.

Fig. 7 shows examples for the *Rotation\** task, where models are asked to generate rotated versions of a LEGO object. Fig. 9 presents *Multiview\** generation cases, illustrating how models interpret viewpoint transformations. Fig. 8 displays *Dependency\** cases, which assess whether models can generate the correct part needed for a given assembly transition.

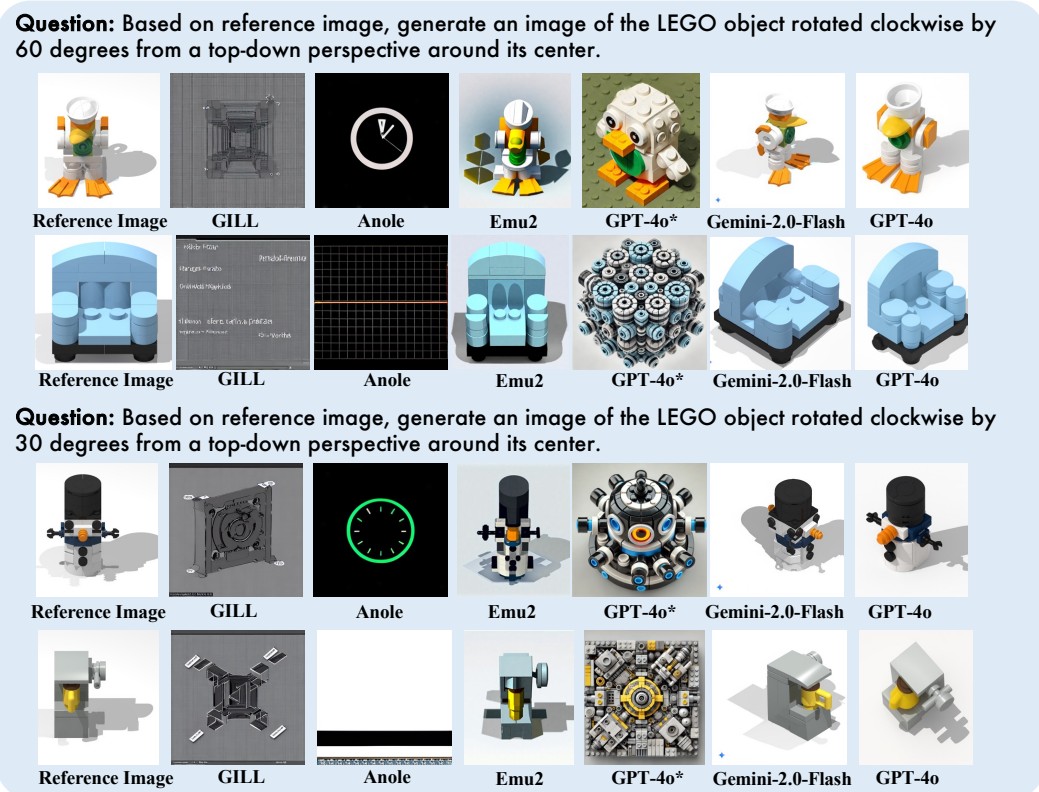

Figure 7: Qualitative visual generation results for Task *Rotation\**. Note: The questions above are slightly simplified for clarity and brevity.

## Dependency*

**Question**: Based on the current assembly state <image 1>. the next assembly state <image 2>, generate an image that shows the LEGO part required to make the transition from the current state to the next state.

Figure 8: Qualitative visual generation results for Task *Dependency*. Note: The questions above are slightly simplified for clarity and brevity.

## Multiview*

**Question:** Based on reference image, generate an image showing the LEGO piece from a top-to-down perspective.

**Question:** Based on reference image, generate an image showing the LEGO piece from a left-to-right perspective.

Figure 9: Qualitative visual generation results for Task *Multiview*. Note: The questions above are slightly simplified for clarity and brevity.

### 8.7 GENERATIVE FULL-STAGE PLANNING EXPERIMENT

Experiments in Sec. 3.2.2 show that current MLLMs struggle severely with multi-step planning even under a simplified discriminative setup. To further test realistic full-stage planning ability, we additionally invesitigate the generative planning ability of current MLLMs. This perspective mirrors how humans naturally perform LEGO assembly: given an initial configuration and a final

target, an intelligent system should be able to synthesize intermediate steps sequentially, effectively demonstrating how to complete the build. Motivated by this, we design an additional experiment that evaluates full-stage, multi-step planning under a realistic multi-turn formulation.

**Experimental setting.** We adopt an interactive, step-by-step generative framework, where the model produces one intermediate state at each turn. Given the initial state $x_0$ and the final target state $x_{k+1}$:

- Turn 1: The model receives $x_0$ and $x_{k+1}$ and is asked to generate the next assembly state $x_1$.
- Turn 2: The model is then asked to generate $x_2$, conditioned on its previously generated output $x_1$.
- Turn (t): At each subsequent turn, the model generates $x_{t+1}$ based on the entire history of generated states.

This process continues until the model attempts to produce $x_k$, yielding a complete generative trajectory from start to goal.

**Models and Evaluation.** We evaluated two current strong models—GPT-5.1 (OpenAI, 2025b) and Gemini-2.5-Flash-Image (Nano-Banana) (Comanici et al., 2025). For each planning horizon (k), we tested 5 cases. Each generated trajectory was assessed by five human annotators. A trajectory is scored as 1 only if the generated intermediate image(s) satisfy the following criteria:

- The model produces an image corresponding to adding exactly one new LEGO piece, consistent with the expected assembly step.
- The newly added piece is placed in the correct location with the correct color.
- All previously existing pieces remain unchanged in color, shape, and position—that is, their overall appearance is preserved as in the previous step.

If any of these conditions is violated, the trajectory receives a score of 0. This criterion ensures that we measure true task completion, rather than coarse visual similarity. Alternative scoring schemes could be adopted depending on downstream applications, but the primary goal here is to assess precise assembly correctness.

We set $k = 1, 2, 3$. Because all models consistently failed, we did not extend evaluation to longer horizons. Although some generated images exhibit reasonable appearance similarity, none of the models succeeded in producing even a single correct intermediate assembly step. The results are shown below, and qualitative examples are shown in Fig. 10, Fig. 11, Fig. 12, Fig. 13, Fig. 14, and Fig. 15.

| k | ChatGPT-5.1 | NanoBanana |
|---|---|---|
| 1 | 0 | 0 |
| 2 | 0 | 0 |
| 3 | 0 | 0 |

**Results: even the strongest current MLLMs fail completely in this setting.** The models cannot produce coherent intermediate assembly states, and errors always emerge immediately at the first step. The results suggest that models lack the ability to maintain 3D structure across turns and cannot perform realistic assembly plans. The results reflects that true generative multi-step planning is far beyond the capability of current MLLMs. Such a task requires not only spatial understanding but also consistent visual imagination, temporal reasoning, and implicit physics which have not yet been fully solved.

The results from LEGO-Puzzles and the generative full-stage planning experiments clearly highlight the substantial gap between current MLLM capabilities and realistic full-stage assembly reasoning. Moving forward, both models and benchmarks will need to evolve beyond static image settings. Future directions include generative multi-step planning, interactive simulation or environment-based evaluations, and dynamic tasks where the model must iteratively propose, verify, and revise intermediate states. Such setups would more closely reflect real-world decision-making and provide a clearer path toward developing MLLMs with genuine spatial reasoning and planning abilities.

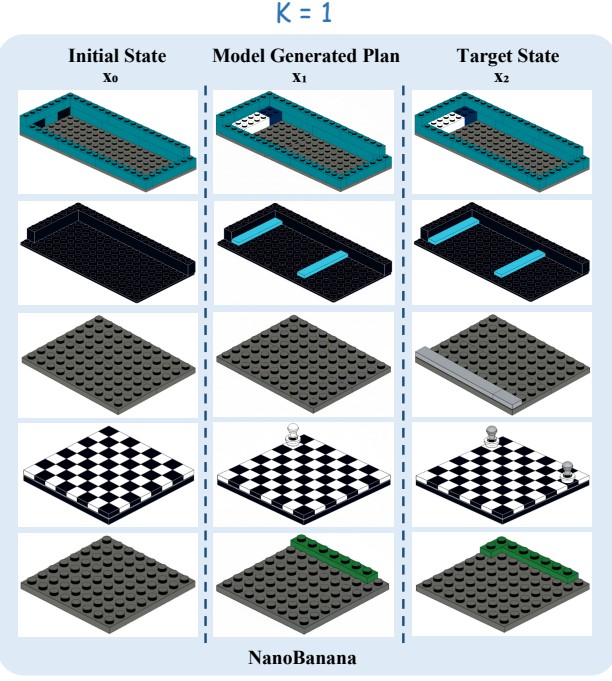

Figure 10: Qualitative visual generation results of ChatGPT-5.1 for generative full-stage planning experiments when k = 1.

Figure 11: Qualitative visual generation results of NanoBanana for generative full-stage planning experiments when k = 1.

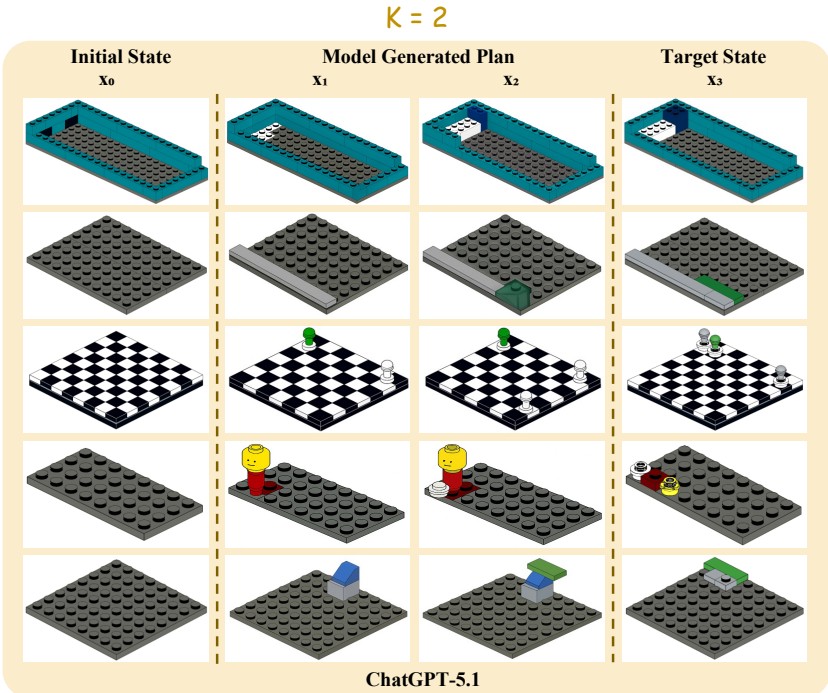

Figure 12: Qualitative visual generation results of ChatGPT-5.1 for generative full-stage planning experiments when k = 2.

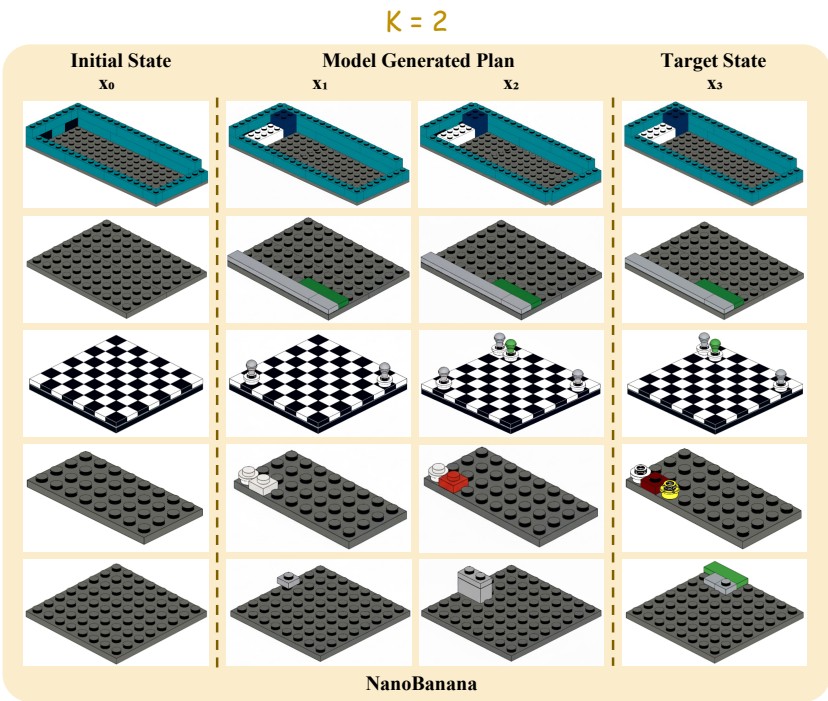

Figure 13: Qualitative visual generation results of NanoBanana for generative full-stage planning experiments when k = 2.

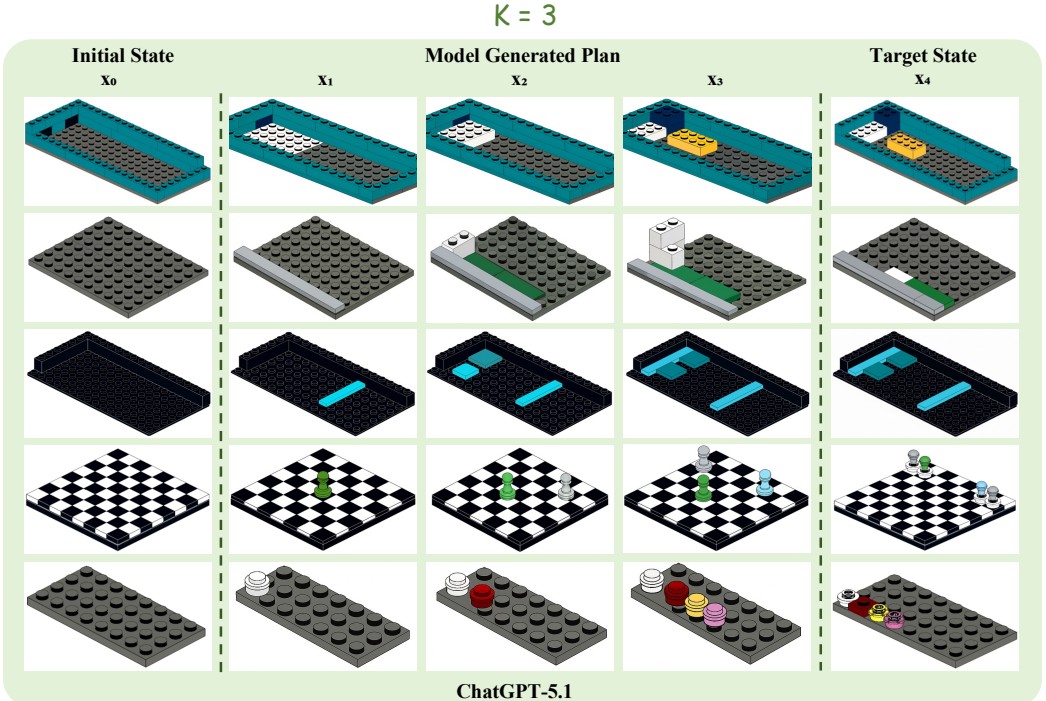

Figure 14: Qualitative visual generation results of ChatGPT-5.1 for generative full-stage planning experiments when k = 2.

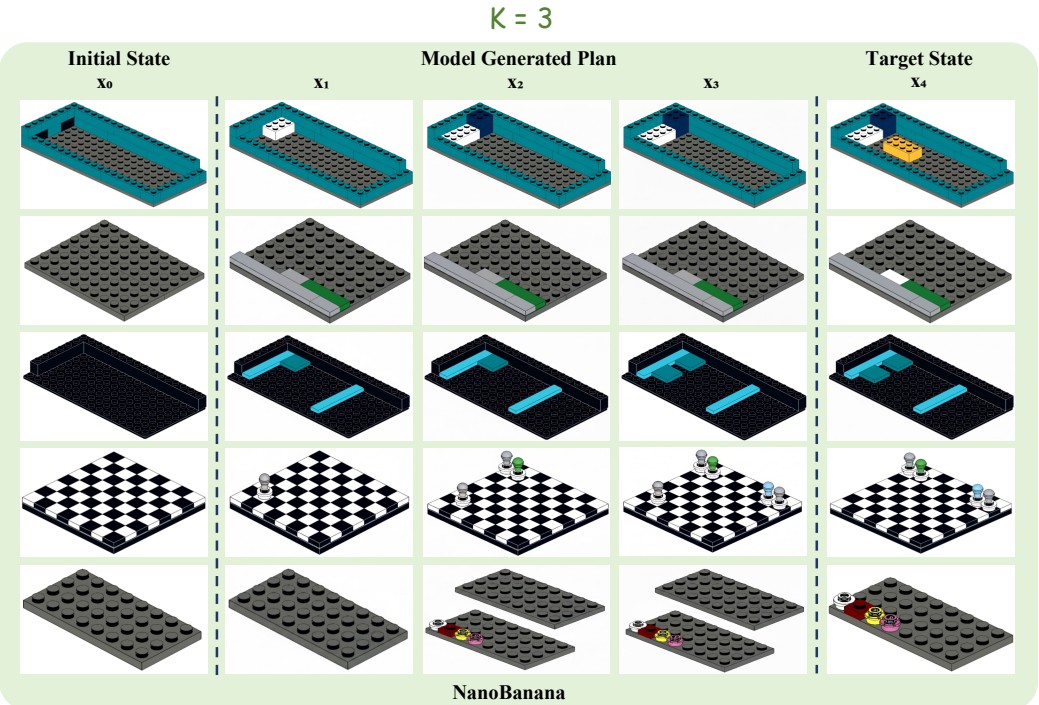

Figure 15: Qualitative visual generation results of NanoBanana for generative full-stage planning experiments when k = 3.

