# OpenReview forum: "LEGO-Puzzles: How Good Are MLLMs at Multi-Step Spatial Reasoning?"
_ICLR.cc/2026/Conference — Submitted to ICLR 2026_

### Official Review · Reviewer_z8xn · 2025-10-31

**Soundness:** 3
**Presentation:** 4
**Contribution:** 3
**Rating:** 8
**Confidence:** 3

**Summary:**

The paper proposes a new benchmark for evaluating the spatial understanding and reasoning capabilities of MLLMs based on LEGO construction. It includes two task sets: an elementary set with 3 levels of spatial reasoning and 11 VQA tasks, and a planning set that extends to multi-step reasoning under noisy conditions. By evaluating over 20 different models, the results highlight limitation in the spatial understand and reasoning capabilities of current MLLMs. The paper also extends the multiple-choice setting to an image-generation task, revealing the limitations of current generative models in instruction-grounded image generation. Moreover, the high correlation between LEGO-Puzzle and 3DSRBench shows that model performance on LEGO-Puzzle (Height and Adjacency) is indicative of real-word spatial reasoning ability.

**Strengths:**

- The paper provides a detailed description of its data curation process, which is a strong point for a benchmark paper. It also includes evaluation prompts and the rubric for human evaluation, both of which enhance transparency.
- The benchmark itself is comprehensive, covering multiple task types and 2 different task sets, which planning to be a more challenging one.
- The evaluation is also comprehensive, encompassing different model families and sizes, as well as both open-source and proprietary models.

**Weaknesses:**

- The image-generation evaluation relies on human experts for scoring, which makes the results less reproducible and harder to verify. While using LLM-as-a-judge might introduce biases, a comparison between human and LLM-based evaluations could be conducted to evaluate the consistency between the two.
- Although the authors show a strong performance correlation on Height and Adjacency tasks between LEGO-Puzzles and 3DSRBench, and I appreciate this analysis, the entire benchmark uses images from a similar visual theme (LEGO construction). Therefore, the generalizability of other task types to real-world scenarios remains somewhat limited.
- The evaluation mainly uses a zero-shot setting. It would benefit from a few-shot comparison and an analysis of prompt sensitivity, especially since many task types involve multiple images. MLLMs could be highly sensitive to the placement of images within the prompt.

**Questions:**

- Based on the examples, it seems that some question types, such as Height and Adjacency, use a single image, while others involve multiple images. This might introduce some confounding factors, it becomes unclear whether performance drops because the model fails to understand spatial relationships or because it struggles to integrate multiple images. For these tasks, should the authors try combining the images into a single composite image to see if performance changes? Alternatively, should the authors analyze how performance varies with the number of images in prompt?
- Could the authors elaborate on the QA construction process? The description in the main text is rather vague, and Appendix 7.4.1 seems to describe the evaluation prompt, which appears unrelated to QA construction.
- For the image-generation evaluation, the paper states that 5 human experts were employed and that scores were given on a scale from 0 to 3 (which seems integer-based according to the rubric). Could the authors clarify how results such as 1.75 were obtained in these cases?

---

> ### Author Response · Authors · 2025-11-22
>
> Dear Reviewer z8xn:
>
> Thank you for your detailed and encouraging feedback, which have greatly improved our work. We appreciate your recognition of the **benchmark’s comprehensive design and evaluation, as well as the transparent presentation of our data curation process, evaluation prompts, and human scoring rubrics.** We also note that your expressed confidence was relatively low, and we hope that our responses below will help strengthen your assessment of our work. We address your concerns in detail below.
>
> > ***Weakness1: Concerns on limited reproducibility due to human-scored image generation,  comparison against LLM-as-a-judge for consistency.***
> >
>
> Thank you for highlighting the potential role of LLM-as-a-judge evaluation in image generation. To explore this direction, we conducted a systematic comparison between human expert scores and LLM-based (GPT-4o) automatic evaluation on image outputs generated by the two strongest models in our benchmark (Gemini-2.0-Flash and GPT-4o) across two evaluation dimensions: Appearance Similarity (App) and Instruction Following (IF).
>
> 1. **We implemented a structured LLM-based evaluation pipeline.** Each generated sample was evaluated by GPT-4o using a detailed rubric mirroring our human annotation protocol. The prompt included the original image(s), the instruction, the model-generated image, and the ground-truth image. GPT-4o assigned 0–3 scores for Appearance Similarity and Instruction Following, following the same criteria used by human experts. We then compared GPT-4o’s scores with human judgments using Pearson correlation, MAE, and MSE.
> 2. **LLM-as-a-judge is not yet suitable to serve as a reliable judge in this setting and exhibits low consistency with human evaluation on strong models.** As shown below, GPT-4o demonstrates substantial disagreement with human ratings when evaluating higher-quality generations from Gemini-2.0-Flash and GPT-4o. For example, Pearson correlation on Appearance Similarity is only 0.133 for Gemini-2.0-Flash and even negative (–0.443) for GPT-4o. MAE and MSE are also large: GPT-4o reaches 0.82 MAE / 0.716 MSE on Appearance Similarity, while Gemini-2.0-Flash shows 1.80 MAE / 3.56 MSE. Deviations remain notable on Instruction Following as well (e.g., 0.77 MAE and 0.762 MSE for Gemini-2.0-Flash). These results suggest that current MLLMs still lack robust spatial understanding of structures and are therefore **not yet suitable to serve as reliable judges** for spatially grounded image-generation tasks that require precise spatial alignment.
>
>
>     | Model | GPT-4o (APP / IF) | Gemini-2.0-Flash (APP / IF) |
>     | --- | --- | --- |
>     | Pearson | -0.443 / 0.298 | 0.133 / 0.685 |
>     | MAE | 0.820 / 0.430 | 1.800 / 0.770 |
>     | MSE | 0.716 / 0.333 | 3.560 / 0.762 |
> 3. **LLM-as-a-judge aligns with human evaluation only on clearly failed cases.** We further evaluated the worst-performing models in our benchmark (GILL and Anole), whose generations are largely unrelated to the input images and instructions (*Section 4* of the paper). As shown below, both human annotators and GPT-4o assigned near-zero scores across both dimensions, indicating that existing MLLMs can reliably identify severe failures but struggle with finer-grained judgments.
>
>
>     | Task | GILL Human (APP / IF) | GILL GPT-4o (APP / IF) | Anole Human (APP / IF) | Anole GPT-4o (APP / IF) |
>     | --- | --- | --- | --- | --- |
>     | rotation | 0.00 / 0.00 | 0.00 / 0.00 | 0.10 / 0.00 | 0.15 / 0.05 |
>     | multi | 0.00 / 0.00 | 0.05 / 0.05 | 0.05 / 0.00 | 0.10 / 0.10 |
>     | position | 0.00 / 0.00 | 0.00 / 0.00 | 0.00 / 0.00 | 0.00 / 0.00 |
>     | dep | 0.00 / 0.00 | 0.00 / 0.00 | 0.00 / 0.00 | 0.00 / 0.00 |
>     | next | 0.00 / 0.00 | 0.00 / 0.00 | 0.00 / 0.00 | 0.00 / 0.00 |
>     | **Average** | **0.00 / 0.00** | **0.01 / 0.01** | **0.03 / 0.00** | **0.05 / 0.03** |
>
> In conclusion, while we agree that reproducible automatic evaluation is desirable, our analysis shows that current LLM-as-a-judge approaches do not yet replicate human judgment on spatially precise, instruction-grounded image-generation tasks. Human evaluation therefore remains essential. Improving spatially aware LLM-based evaluators is an important direction for future work.

---

> > ### Author Response · Authors · 2025-11-22
> >
> > > ***Weakness 2: Concerns on limited generalizability to real-world scenarios.***
> > >
> >
> > Thank you for raising this important concern. We acknowledge that LEGO-Puzzles uses images within a consistent LEGO construction theme. However, **the choice of LEGO as the visual domain is deliberate and well-motivated.**
> >
> > 1. **LEGO-Puzzles leverages the LEGO assembly process as a natural environment for evaluating spatial understanding and sequential reasoning.** The assembly process of a complete LEGO model consists of *discrete, interpretable, and visually consistent* construction steps, each requiring precise understanding of 3D spatial relationships and forming a coherent step-by-step progression. This property allows us to systematically examine spatial reasoning across well-defined transitions, which would be **extremely difficult to achieve with unconstrained natural images.** *This makes LEGO a uniquely effective medium for studying multi-step spatial reasoning in a controlled yet meaningful setting.*
> > 2. **Constructing more complex tasks in natural-image settings is impractical, LEGO-Puzzles makes them feasible.** Beyond *Height* and *Adjacency*, tasks such as next-step prediction and multi-step planning are extremely challenging to construct reliably with natural images due to uncontrolled viewpoint changes, lack of stepwise ground truth, and the difficulty of obtaining precise intermediate states. In contrast, LEGO-Puzzles enables us to build these complex tasks under strict control while maintaining physical realism. We therefore believe that the strong natural-image correlation observed on *Height* and *Adjacency* extends to more complex tasks, and that LEGO-Puzzles serves as a meaningful and scalable proxy for real-world spatial reasoning and planning.
> > 3. **LEGO-Puzzles is grounded in real, physically valid LEGO projects rather than arbitrary synthetic scenes.** All scenes in LEGO-Puzzles are derived from real LEGO projects and official building instructions, ensuring that every object and assembly step corresponds to an authentic, physically feasible structure. As a result, the spatial transformations in our benchmark closely mirror those found in real-world assembly tasks. This realism is further supported by the strong performance correlation with 3DSRBench (*Section 5, Table 4* in paper), which is based on natural images. *These properties collectively mitigate concerns that the benchmark is detached from real-world spatial reasoning.*
> >
> > In conclusion, LEGO-Puzzles achieves a balance between realism and experimental control, making it an ideal testbed for evaluating both spatial understanding and multi-step sequential reasoning in MLLMs.

---

> > > ### Author Response · Authors · 2025-11-22
> > >
> > > > ***Weakness 3: Lack of few-shot and prompt-sensitivity analysis.***
> > > >
> > >
> > > Thank you for the insightful suggestion. To evaluate the effects of few-shot prompting and the sensitivity to image placement, we conducted the following experiments and analysis.
> > >
> > > 1. **Few-shot prompting leads to consistent performance improvements.** To assess how few-shot prompting influences model performance, we conducted 1/3/5-shot experiments using Qwen3-VL-8B-Instruct on four Spatial Understanding tasks: *Height, Adjacency, Rotation,* and *Multi-View*. For each task, we randomly selected 5 examples as demonstrations and evaluated the remaining 95 samples. The results are shown below:
> > >
> > >
> > >     | number-shot | Height | Adjacency | Rotation | Multi-View |
> > >     | --- | --- | --- | --- | --- |
> > >     | 0-shot | 32.63% | 62.11% | 27.37% | 46.32% |
> > >     | 1-shot | 29.47% | 66.32% | 28.42% | 50.53% |
> > >     | 3-shot | 41.05% | 66.32% | 33.68% | 48.42% |
> > >     | 5-shot | 42.11% | 68.42% | 32.63% | 48.42% |
> > >
> > >     We observe that the model shows consistent improvement under few-shot prompting. The slight fluctuations in the 1-shot Height result are expected, as both 0-shot and 1-shot performance remain close to random guessing. Overall, these results indicate that few-shot prompting can partially enhance spatial reasoning ability, and it is important to note that even with such demonstrations the model’s performance remains far below human accuracy.
> > >
> > > 2. **Prompt sensitivity: model performance shows low sensitivity to image placement.** For single-image tasks (*Height* and *Adjacency*), we tested whether image placement in the prompt affects performance by moving the image before or after the question using Qwen3-VL-8B-Instruct. The model’s accuracy remained essentially unchanged, indicating low sensitivity to image placement in these settings. For multi-image tasks, we merge all images into a single composite image and evaluate the performance of the model(details provided in W3). The results show that performance only drops when the composite image becomes visually overloaded (six or more images in our setting), suggesting that the limitation comes from visual complexity, not from the ordering or placement of images in the prompt.
> > >
> > >
> > >     | Acc | Height | Adjacency |
> > >     | --- | --- | --- |
> > >     | **Baseline** | 34.0% | 64.0% |
> > >     | **Image-Position-Changed** | 31.0% | 63.0% |

---

> ### Author Response · Authors · 2025-11-22
>
> > ***Question 1: Need tests with composite images or varying image counts.***
> >
>
> Thank you for the thoughtful question. We would like to clarify that performance differences in LEGO-Puzzles are driven primarily by the *intrinsic difficulty of each task*, rather than by the number of images included in the prompt. To verify this, we conducted the following experiments and analysis.
>
> - **Experiments: combining multi-images into a single composite image.** To directly test whether multi-image prompts introduce confounding factors, we conducted an additional experiment using Qwen3-VL-8B-Instruct, **where we **merged all images in each task into a single composite image**. We applied this to several tasks covering a range of image counts: *Rotation* (2 images), Rotation_Status* (3), *Multiview* (5), Dependency* (6), *Position* (7), and N*ext_Step* (8). Results are shown below:
>
> | Task | Rotation (2) | rotation_status (3) | Multiview (5) | dependency (6) | position (7) | next_step (8) |
> | --- | --- | --- | --- | --- | --- | --- |
> | **Multiple Images** | 29.0% | 54.0% | 44.0% | 80.0% | 38.0% | 57.0% |
> | **Single Composite Image** | 29.0% | 58.0% | 44.0% | 58.0% | 23.0% | 39.0% |
> - **Results: performance begins to drop substantially when six or more images are merged into a single composite image**. This indicates that the degradation comes **not** from using multiple images in the prompt. In fact, using separate images is often *more helpful*, because each image is visually clean, clearly framed, and semantically unambiguous. By contrast, when many images are compressed into a single image, the image becomes visually dense and overloaded. This may exceeding the visual processing capacity of current MLLM encoders, which leads to lower accuracy.
>
> > ***Question 2: Unclear QA construction process.***
> >
>
> Thank you for pointing this out. The description in the main text may not fully convey how the QA pairs are actually constructed. The evaluation prompt shown in *Appendix 8.4.1* is part of the QA construction process. Below we provide a clearer explanation of our QA generation process, and we have revised the paper accordingly by adding a dedicated section in *Appendix 8.5*.
>
> 1. **Our QA construction is fully template-based and task-specific.** For each task, we manually design a template that specifies:
>     - Question
>         - the instruction defining the model’s role (e.g., “You are an expert in LEGO assembly.”),
>         - how images are referenced (e.g., using tokens like <image 1>),
>         - the precise question format required for that task,
>     - Answer: the ground-truth answer derived from the labeled assembly metadata.
> 2. **Ground-truth answers are generated directly from the structured assembly data.** Since each LEGO project is constructed step-by-step with precise metadata (e.g., part ID, assembly-step index), the correct answer for each question thus can be automatically extracted. For example, in the *Next-Step* task, given the current assembly-step index *k* and the final target assembly-step index *m*, the ground-truth next state is simply the image corresponding to assembly-step index *k + 1*. This provides a clear and deterministic answer.
>
> > ***Question3: Unclear how fractional image-generation scores were derived from integer-based human ratings.***
> >
>
> Thank you for the question. We now clarify this in detail. Although each individual rating is an integer from 0 to 3 and each image is evaluated by 5 human experts, **each task contains 20 evaluation cases**. The reported score is the average over all annotators and all 20 cases, so aggregating these integer ratings naturally yields fractional values such as 1.75.

---

> ### Author Response · Authors · 2025-11-27
> **looking forward to your reply**
>
> Dear Reviewer z8xn,
>
> We sincerely appreciate the time, effort, and thoughtful attention you have devoted to reviewing our paper. Your constructive feedback and valuable suggestions have been instrumental in helping us improve our work.
>
> As the author–reviewer discussion period is approaching its end, we wanted to ensure that our responses have fully addressed your concerns, including:
> - **analysis of the image-generation evaluation comparing human and LLM judging,**
> - **discussion of the LEGO-Puzzles's generalizability and the rationale behind our design choices,**
> - **experiments on few-shot settings and prompt sensitivity,**
> - **analysis of the performance impact of multi-image inputs vs. combining them into a single composite image**
> - **clarification of the QA construction process,**
> - **clarification of the image-generation scoring procedure.**
>
> If there are still points that you feel need further explanation or discussion, please let us know. We would be more than happy to provide additional details or clarification, and we value the opportunity to address any remaining questions before the discussion phase concludes.
>
> Once again, thank you for your dedication, time, and insights. We deeply value your contribution to the review process and look forward to hearing your thoughts.
>
> Sincerely,
>
> LEGO-Puzzles Authors

---

### Official Review · Reviewer_2tpf · 2025-10-31

**Soundness:** 3
**Presentation:** 3
**Contribution:** 3
**Rating:** 4
**Confidence:** 5

**Summary:**

This paper introduces LEGO-Puzzles, a new benchmark designed to evaluate the multi-step spatial reasoning capabilities of MLLMs. The benchmark comprises two main task sets: an "Elementary" set of 11 VQA tasks (1,100 samples) that test skills from basic spatial understanding to multi-step sequential logic , and a "Planning" set (PLAN-k-Step) that requires models to generate assembly plans of varying lengths (k=1 to 8). A comprehensive evaluation of 23 MLLMs reveals critical limitations , showing that even top proprietary models lag human performance by over 20% and that planning accuracy degrades to 0% as sequence length increases. Furthermore, the study demonstrates that MLLMs' reasoning abilities do not transfer to image generation, where performance is near-zero.

**Strengths:**

* This paper is clear writing and easy to follow.

* The paper effectively identifies and addresses a significant gap in current MLLM evaluation: multi-step, sequential spatial reasoning. This capability is a clear prerequisite for downstream applications such as embodied models, and mm agentic models, yet it remains largely unexplored by existing benchmarks that focus on static VQA.

* The evaluation is extensive, testing 23 SOTA MLLMs, including both leading proprietary and open-source models. This breadth provides a convincing and broad snapshot of the field's current limitations, showing this is a universal, not model-specific, failure.

**Weaknesses:**

* The "Planning" task , while effective, is still fundamentally a discriminative task: models must select and order from a given set of candidate images. This is a significant simplification of true generative planning, which would require the model to generate a plan from scratch. A more robust and revealing benchmark would necessitate a generative paradigm, where the model must produce the multi-step plan from scratch, rather than simply selecting and sorting provided options. Furthermore, the paper could benefit from a discussion on future work moving beyond static datasets entirely.

* The paper's reporting on human performance is confusing. The abstract and Planning set results  state humans "solve all the tasks perfectly" (100%). However, Table 2, which evaluates the "Elementary" set, shows human proficiency at 93.6%, with scores as low as 70% on the 'Height' task. It is unclear why human experts would struggle with basic elementary tasks but achieve perfection on the much more complex multi-step planning tasks. Are there any typos or mistakes?

* The benchmark's emphasis on multi-step, goal-oriented assembly makes it an excellent candidate for evaluating agentic systems, not just standard VQA models. A notable omission in the current evaluation is the absence of baselines from general-purpose VLA models and mm agentic model. This would allow for a crucial discussion on the differing capabilities of static MLLMs versus agentic frameworks in solving complex, sequential spatial reasoning tasks.

**Questions:**

* Could you please clarify the discrepancy in human performance? Why did the 30 human experts score relatively low (e.g., 70% on 'Height', 93.6% overall) on the "Lite" Elementary set , while the five experts in the Planning set achieved 100% on all k steps?

* The finding that MLLMs default to 2D projections for 3D tasks like 'Height' is insightful. Do you experiment with any prompting strategies to mitigate this?

* Can you evaluate the difference between multimodal reasoning models and chat models?

* Can this task be structured into an RLVR environment, and provide rewards; and how to make it scalable?

---

> ### Author Response · Authors · 2025-11-22
>
> Dear Reviewer 2tpf:
>
> Thank you for your encouraging and detailed comments, which have greatly improved our work. We appreciate you find our work **effectively identifies and addresses a significant gap in current MLLM evaluation: multi-step.** We address your concerns in detail below:
> > ***[1/3] Weakness 1: Concerns on simplification of planning to a discriminative task instead of true generative planning.***
> >
>
> Thank you for the thoughtful suggestion. We agree that moving toward generative planning is an important direction. Below we clarify why we adopt the current discriminative formulation, and **we also present new experiments that explore generative multi-step planning.**
>
> 1. **Why we adopt a discriminative “Plan-k-Step” formulation instead of full generative planning.** Directly asking an MLLM to output a *full* assembly plan from scratch is currently infeasible for several practical reasons:
>     - **Ambiguity in defining a generative plan** **without a well-defined coordinate system.** To generate a true plan, the model would need to output where exactly to place each LEGO piece and how to orient it in 3D space. That requires a clear coordinate system (e.g., x/y/z axes, rotation angles). But in real LEGO assembly or robot assembly, such coordinate systems are not naturally defined: humans do not place LEGO pieces using absolute coordinates and robots rely on hardware, camera calibration, arm-specific kinematics, and workspace definitions. If we artificially impose a coordinate system just for the benchmark, the resulting task becomes detached from how assembly actually works in the real physical world. It also introduces many arbitrary design choices that would unfairly influence model evaluation.
>     - **MLLMs are not yet able to generate multiple intermediate images.** One option for a generative plan is to have the model directly generate the next assembled state as an image. However, current MLLMs cannot reliably output multiple images conditioned on an image sequence. This makes end-to-end generative assembly fundamentally out of reach for today’s models.
>
>     Given these constraints, our “Plan-k-Step” setting provides a **practical and controlled** middle ground: it removes the need for coordinate engineering while still revealing whether models can reason consistently across multiple steps. Importantly, *Section 3.3.2* already shows that **even with the simplified select-and-order interface, all models fail when k reaches 8**, with accuracy dropping to zero. This indicates that current MLLMs struggle severely with multi-step planning even under the simplified discriminative setup.

---

> ### Author Response · Authors · 2025-11-22
>
> > ***[2/3] Weakness 1: Concerns on simplification of planning to a discriminative task instead of true generative planning.***
> >
> 2. **Experiments on Generative Full-Stage LEGO Planning.** Experiments in *Section 3.3.2* already show that current MLLMs struggle severely with multi-step planning even under a simplified discriminative setup. To further evaluate the realistic full-stage planning ability of MLLMs, we additionally designed a generative full-stage planning experiment. This perspective mirrors how humans naturally perform LEGO assembly: given the current state and the final goal, we expect a system to generate each subsequent intermediate state step by step, which effectively teaches us how to complete the build. Motivated by this perspective, we conducted a new experiment to directly evaluate generative multi-step planning from scratch under a realistic multi-turn formulation.
>
>     **Experimental setting.** We evaluate generative planning in an interactive multi-turn setting, where the model generates one intermediate state at a time. Given an initial state $x_0$ and the final target state $x_{k+1}$:
>
>     1. Turn 1: We provide $x_0$ and $x_{k+1}$ and ask the model to generate the next assembly state $x_1$.
>     2. Turn 2: We then ask the model to generate $x_2$, based on the previously generated $x_1$.
>     3. Subsequent turns: At each turn $t$, the model is asked to generate the next state $x_{t+1}$ given all earlier generated states.
>
>     This iterative process continues until the model attempts to produce $x_k$, forming the full multi-step trajectory.
>
>     **Models and Evaluation.** We evaluate two frontier models—GPT-5.1 and NanoBanana. For each planning horizon $k$, we tested 5 cases (limited by the lack of multi-image generation API), but this scale is sufficient to reveal the core failure patterns (see results below). Each generated trajectory was assessed by five human annotators. A trajectory is scored as 1 only if the generated intermediate image(s) satisfy the following criteria:
>
>     - The model produces an image corresponding to adding exactly one new LEGO piece, consistent with the expected assembly step.
>     - The newly added piece is placed in the correct location with the correct color.
>     - All previously existing pieces remain unchanged in color, shape, and position—that is, their overall appearance is preserved as in the previous step.
>
>     If any of these conditions is violated, the trajectory receives a score of 0. This criterion ensures that we measure true task completion, rather than coarse visual similarity. Alternative scoring schemes could be adopted depending on downstream applications, but the primary goal here is to assess precise assembly correctness.

---

> ### Author Response · Authors · 2025-11-22
>
> > ***[3/3] Weakness1: Concerns on simplification of planning to a discriminative task instead of true generative planning.***
> >
> We set k = 1, 2, 3. Since all models consistently failed at these horizons, we did not extend the evaluation to longer sequences. Although some generated images exhibit reasonable *appearance similarity*, none of the models succeeded in producing even a single correct intermediate assembly step. The results are shown below, and more qualitative examples can be found in *Appendix 8.7*:
>
> | **k** | **ChatGPT-5.1** | **Nano-Banana** |
> |:---:|:---:|:---:|
> | 1 | 0 | 0 |
> | 2 | 0 | 0 |
> | 3 | 0 | 0 |
>
> **Results: even the strongest current MLLMs fail completely in this setting.** The models cannot produce coherent intermediate assembly states, and errors always emerge immediately at the first step, such as placing the newly added piece in an incorrect location, altering the color or shape of the pieces, or misunderstanding the instruction by adding multiple pieces at once.The results suggest that models lack the ability to maintain 3D structure across turns and cannot simulate realistic assembly transitions. These observations reflect that **true generative multi-step planning remains far beyond** the capability of current MLLMs. This **supports the design choice behind the *PLAN-K-Step* task of LEGO-Puzzles**: it strikes a practical balance—challenging enough to expose concrete weaknesses in spatial reasoning and multi-step planning, yet not so difficult that all models fail entirely. In contrast, full-stage generative planning requires a much broader set of abilities such as consistent visual imagination, temporal reasoning, and implicit physics, which have not yet been fully solved.
>
> 3. **Discussion on future work moving beyond static datasets entirely.** We agree with you that it is important to discuss future work moving beyond static datasets entirely. We have added this discussion in *Appendix 8.7* of the revised paper. We summarize the key points here. The results from LEGO-Puzzles and the new generative experiment clearly highlight the substantial gap between current MLLM capabilities and realistic full-stage assembly. Moving forward, both models and benchmarks need to go beyond static image settings such as enabling models to generate each step of a plan and interact with a simulated environment. Such setups would more closely reflect real-world decision-making and provide a path toward developing MLLMs with more reliable planning abilities.

---

> ### Author Response · Authors · 2025-11-22
>
> > ***Weakness2 & Question1: Concerns on inconsistent human performance reporting.***
> >
>
> Thank you for pointing out this. The difference in human performance between the Elementary set and the Planning set is expected and can be explained as follows.
>
> 1. **Human errors in the Elementary set mainly arise from deliberate 2D–3D visual ambiguities.** In the *Height* task of the Elementary set, many items are intentionally designed to include intentional 2D–3D ambiguities that can lead to occasional human mistakes. In these cases, humans may rely on quick visual intuition rather than careful spatial reasoning—especially when the blocks are not adjacent and the height must be inferred through spatial imagination and reasoning rather than direct comparison. When objects are farther apart or partially occluded, people may overlook subtle depth differences and misjudge the true height. As a result, human accuracy in the Elementary set is slightly lower, but it is still far above the performance of current MLLMs. Several such cases are illustrated in Appendix 8.2. We also found that these mistakes largely come from not inspecting the images carefully; when participants were explicitly reminded to pay closer attention, they achieved 100% accuracy.
> 2. **The Planning set is more aligned with natural human spatial reasoning ability, which explains the perfect human performance.** Unlike the Elementary set, the Planning tasks do not require such fine height estimation. Participants are asked to identify differences across assembly steps such as color changes and part placement shifts. These cues are concrete and physically grounded. As a result, these tasks do not easily lead to careless errors. Humans can therefore solve them almost perfectly, even though the tasks appear more “complex” in terms of multi-step structure. However, these same tasks remain highly challenging for current MLLMs. This contrast highlights the substantial gap between human spatial reasoning ability and the capabilities of existing models.
>
> > ***Weakness3: Concerns on missing evaluation with VLA and multimodal agentic models.***
> >
>
> Thank you for raising this important point. We agree that the multi-step, goal-oriented nature of LEGO-Puzzles makes it a natural candidate for evaluating agentic systems, not just standard VQA-style MLLMs, But we would like to clarify the specific scope and design philosophy of our work regarding this point:
>
> 1. **Applying LEGO-Puzzles directly to agentic systems requires a fully interactive environment, which is non-trivial.** We agree that it would be very interesting to study how spatial reasoning capabilities measured by LEGO-Puzzles translate into agentic interaction abilities. However, evaluating agentic models requires constructing an interactive assembly environment—typically involving a physics engine, simulatable LEGO parts, and a continuous action space. At present, LEGO-Puzzles is designed as a static benchmark and does not include such an environment. Similar benchmarking efforts such as 3DSRBench[1], SpatialRGPT[2], and SpatialEval[3]—also focus on evaluating the spatial reasoning ability of MLLMs without introducing agentic interaction. This further highlights that extending LEGO-Puzzles to an interactive agent setting is non-trivial and represents an open direction for future research. We hope that LEGO-Puzzles can provide a foundation and motivation for such developments.
> 2. **Our evaluation focuses on foundational multi-step spatial reasoning, not action generation.** LEGO-Puzzles aims to evaluate the spatial reasoning ability of *foundation MLLMs* under a controlled VQA-style setting. It measures the spatial understanding and sequential reasoning abilities of MLLMs where no interaction is involved. This is conceptually different from evaluating VLA or agentic models, which must predict actions, manipulate an environment, or perform sequential decision-making. Applying LEGO-Puzzles directly to such systems would require re-formulating the tasks into an interactive control problem, which goes beyond the scope of this work.
> 3. **Nevertheless, LEGO-Puzzles provides a valuable foundation for future VLA/agentic evaluation.** Many recent VLA or agentic systems—such as UI-TARS, Magma, and related models—are finetuned on top of general-purpose VLMs. Their performance is fundamentally constrained by the underlying model’s spatial understanding. By offering a clean and controlled way to measure multi-step spatial reasoning, LEGO-Puzzles can help diagnose baseline weaknesses in the underlying MLLMs before moving to interactive settings. In this sense, our benchmark provides a useful building block for future research on agentic multi-step planning.
>
> [1] 3DSRBench: A comprehensive 3d spatial reasoning benchmark.
>
> [2] SpatialRGPT: Grounded spatial reasoning in vision-language models. NeurIPS,2024.
>
> [3] Is a picture worth a thousand words? delving into spatial reasoning for vision language models. NeurIPS,2024.

---

> > ### Author Response · Authors · 2025-11-22
> >
> > > ***Question2: Prompting strategies for mitigating 2D projection bias in 3D tasks.***
> > >
> >
> > Thank you for the question. Yes, we experimented with several prompting strategies to mitigate the issue. **We tested four types of prompts** on the *Height* task using Qwen3-VL-8B-Instruct:
> >
> > - Explicitly stating that the task requires 3D reasoning (W Explicit 3D),
> > - Omitting any explicit mention of 3D (W/O Explicit 3D),
> > - Chain-of-Thought prompting (CoT),
> > - Few-shot prompting (1/3/5-shot)—For each task, we randomly sampled 5 examples as demonstrations and evaluated the remaining 95 test samples.
> >
> >
> > Below we show the results:
> >
> > 1. **W Explicit 3D vs. W/O Explicit 3D**
> >
> >
> >     | Setting | Height |
> >     | --- | --- |
> >     | W Explicit 3D | 34.0% |
> >     | W/O Explicit 3D | 35.0% |
> >
> >     **Explicitly stating that the environment is 3D does *not* change the outcome.** We added clear instructions in the prompt explicitly stating that the environment is 3D and that the model should reason about actual height in 3D space. However, the performance with and without 3D instructions (34.0% vs. 35.0%) is almost identical. This indicates that simple prompt-based cues are not sufficient to shift the model away from its default 2D heuristic.
> >
> > 2. **Chain-of-Thought prompting.**
> >
> >
> >     | Setting | Height |
> >     | --- | --- |
> >     | CoT | 42.0% |
> >
> >     **Chain-of-Thought prompting yields a moderate but still limited improvement.** Encouraging the model to reason step by step increases its accuracy on the *Height* task to 42.0%. However, the gain remains modest, merely pushing the model slightly beyond random guessing.
> >
> > 3. **Few-shot prompting**
> >
> >
> >     | number-shot | Height |
> >     | --- | --- |
> >     | 0-shot | 32.6% |
> >     | 1-shot | 29.5% |
> >     | 3-shot | 41.1% |
> >     | 5-shot | 42.1% |
> >
> >     Few-shot prompting leads to improvement in the 3-shot and 5-shot settings. The fluctuation observed in the 1-shot result is expected, as both 0-shot and 1-shot performance remain close to random guessing. Overall, the improvement is still limited: even with demonstrations, the model only marginally surpasses the random-guessing baseline, and its underlying failure mode remains unchanged.
> >
> >
> > These results show that current MLLMs still rely heavily on a 2D perspective when interpreting scenes. This likely reflects their pretraining distribution: most training images may come from 2D natural photographs or web imagery, with very limited supervision that requires explicit 3D reconstruction. Consequently, the model tends to rely on 2D visual shortcuts.

---

> > > ### Author Response · Authors · 2025-11-22
> > >
> > > > ***Question 3: Evaluation on the difference between multimodal reasoning models and chat models?***
> > > >
> > >
> > > Thank you for the question. To evaluate the differences between multimodal reasoning-oriented models and standard chat-oriented models, we compared Qwen3-VL-8B-Instruct (chat model) with Qwen3-VL-8B-Thinking (reasoning-enhanced model). We selected these two models because they belong to the same model family and differ primarily in whether they include reasoning-oriented fine-tuning. This makes them an ideal pair for isolating the effect of “reasoning mode” on spatial reasoning performance. The results are shown below.
> > >
> > > | **Task** | **Spatial Understanding** |  |  |  | **Single-Step Reasoning** |  |  | **Multi-Step Reasoning** |  |  |  |  |
> > > | --- | --- | --- | --- | --- | --- | --- | --- | --- | --- | --- | --- | --- |
> > > | Models | Height | Adjacency | Rotation | Multiview | Next-Step | Dependency | Rotation Stat. | Position | Backwards | Ordering | Outlier | Avg |
> > > | Qwen3-VL-8B-Instruct | 34.0% | 64.0% | 30.0% | 43.0% | 55.0% | 79.0% | 58.0% | 33.0% | 45.0% | 14.0% | 30.0% | 44.1% |
> > > | Qwen3-VL-8B-Thinking | 36.0% | 56.0% | 31.0% | 48.0% | 40.0% | 77.0% | 50.0% | 35.0% | 54.0% | 32.0% | 22.0% | 43.7% |
> > >
> > > We find that:
> > >
> > > 1. **Reasoning models do not outperform chat models on spatial reasoning.** The Thinking model shows only marginal improvements on tasks such as *Height*, *Rotation*, and *Multiview*, and even drops significantly on others such as *Adjacency*, *Rotation Status*, and *Outlier*. This indicates that stronger chain-of-thought or symbolic reasoning does not directly translate into better spatial perception or 3D structural understanding.
> > > 2. **Overall performance between the two models is nearly identical.** The average accuracy differs by only **0.4%**, suggesting that multimodal spatial reasoning is not bottlenecked by logical reasoning capability, but rather by other factors such as visual representation quality and 3D spatial perception abilities of the model. This highlights that improving spatial reasoning cannot be achieved simply by scaling reasoning steps. Instead, it points to a clear direction for future model development: strengthening the visual and spatial foundations that support higher-level reasoning.
> > >
> > > > ***Question 4: Structuring the task into an RLVR environment with reward design.***
> > > >
> > >
> > > Thank you for the question.
> > >
> > > 1. **LEGO-Puzzles can be structured into an RLVR environment with verifiable rewards.** Each task in LEGO-Puzzles provides ground-truth answers, so a model’s prediction can be directly compared with the correct output to compute rewards. Moreover, the sequential nature of LEGO assembly naturally lends itself to an RLVR formulation: each intermediate step is well-defined, and the model’s predicted next state can be evaluated against the ground-truth next image. This makes LEGO-Puzzles inherently compatible with reward-driven learning frameworks where spatial reasoning is assessed in a step-by-step manner.
> > > 2. **Regarding scalability**, this depends on the desired level of interaction:
> > >     - **For the current static VQA-style setting**, the existing tasks can be used directly for RLVR training since each prediction can be grounded and rewarded immediately.
> > >     - **To scale toward an agent-based interactive environment**, additional components are required. In particular, a simulation engine is necessary for continuous interaction. To move beyond static image comparison, the environment must simulate realistic LEGO assembly dynamics such as verifying whether pieces are correctly attached and enforcing stability constraints. The simulator must also be capable of generating new intermediate states in response to the agent’s actions.

---

> ### Author Response · Authors · 2025-11-27
> **looking forward to your reply**
>
> Dear Reviewer 2tpf,
>
> We sincerely appreciate the time, effort, and thoughtful attention you have devoted to reviewing our paper. Your constructive feedback and valuable suggestions have been instrumental in helping us improve our work.
>
> As the author–reviewer discussion period is approaching its end, we wanted to ensure that our responses have fully addressed your concerns, including:
> - **extending the evaluation beyond the discriminative planning formulation toward more generative planning settings,**
> - **clarifying the reporting of human performance across different subsets of LEGO-Puzzles,**
> - **analysis of the potential of using LEGO-Puzzles to evaluate VLA systems,**
> - **conducting additional experiments with prompting strategies to mitigate 2D projection bias in 3D tasks,**
> - **providing further comparisons between multimodal reasoning models and chat-centric models, and**
> - **offering more analysis and discussion on structuring LEGO-Puzzles into an RLVR environment with verifiable rewards.**
>
> If there are still points that you feel need further explanation or discussion, please let us know. We would be more than happy to provide additional details or clarification, and we value the opportunity to address any remaining questions before the discussion phase concludes.
>
> Once again, thank you for your dedication, time, and insights. We deeply value your contribution to the review process and look forward to hearing your thoughts.
>
> Sincerely,
>
> LEGO-Puzzles Authors

---

> > ### Comment · Reviewer_2tpf · 2025-11-28
> >
> > It would be beneficial to briefly discuss the implementation details for building similar assembly-based tasks, as having an interactive environment is of greater importance for agentic systems. However, since the authors have resolved the majority of my concerns, I intend to increase my rating.

---

> > > ### Author Response · Authors · 2025-12-01
> > > **Appreciating the Reviewer’s Positive Feedback and Further Discussing the Interactive Environment Implementation**
> > >
> > > We sincerely thank you for your thoughtful feedback. Your comments have greatly helped us improve our work. **We are grateful that you feel your major concerns have been addressed and that you intend to increase your rating.** To respond to your suggestion regarding the implementation details for building interactive assembly-based tasks, particularly for agentic systems, we provide the following discussion. Building on the data-generation pipeline described in *Section 2.2* of our paper, we outline a feasible implementation roadmap for such an interactive system across three key dimensions:
> > >
> > > 1. **Action Space: Designing Executable, Atomic Operations.** Instead of free-form text responses, an agent would operate through structured function calls. The discrete assembly operations defined in our benchmark naturally correspond to executable APIs, such as SelectPart(part_id), Rotate(orientation), Place(x, y, z), and Attach(target_part_id, connection_point). These atomic actions define a clean and verifiable action space. They also serve as the bridge between high-level reasoning (VLMs) and low-level physical manipulation (environment), enabling precise control over the assembly process.
> > > 2. **Environment: Automating the Execution–Rendering–Observation Loop.** An interactive setting requires an environment capable of executing actions and returning updated observations. A typical loop operates as follows:
> > >     - **Execution.** When the LLM agent outputs an action, the environment parses the parameters and the updates the structured assembly file (e.g., the .io format used in our pipeline), which stores all part-level metadata, including part IDs, placement coordinates, and orientations.
> > >     - **Automated Rendering.** After updating the structured assembly file, the environment triggers the rendering engine (such as the LEGO Studio software used in *Section 2.2* of our paper) to generate the corresponding visual output.
> > >     - **Observation.** The rendered image is returned to the agent as the next observation, enabling it to produce the subsequent action.
> > >
> > >     This multi-turn loop continues until the agent deliberately issues a *STOP* action, completing the episode.
> > >
> > > 3. **Verifiable Reward Design: deterministic state-based evaluation.** Ground-truth assembly trajectories in LEGO-Puzzles provide a straightforward basis for reward computation. Since the environment maintains a structured assembly file, we can directly evaluate each step by checking whether the agent’s output matches the ground truth in terms of part IDs, discrete (x, y, z) placements, and orientations. If the predicted assembly state matches the ground truth exactly, the agent receives a reward of 1; otherwise, the reward is 0. This deterministic state-based evaluation ensures a clear, reliable, and unambiguous reward signal.
> > >
> > > In Conclusion, such an interactive environment would be a natural extension of LEGO-Puzzles, enabling grounded multi-step planning, spatial manipulation, and agent-based decision-making in more realistic settings. We appreciate your suggestion and believe this direction will be valuable for future research in multimodal and agentic systems.

---

### Official Review · Reviewer_B7zR · 2025-10-31

**Soundness:** 3
**Presentation:** 3
**Contribution:** 2
**Rating:** 6
**Confidence:** 4

**Summary:**

This paper introduces a novel benchmark, LEGO-Puzzles, focusing on the MLLM multi-step spatial reasoning capabilities. LEGO-Puzzles include two sets of questions: (1) Elementary Set: Multiple choice questions which test MLLM’s different capabilities, such as spatial understanding, and planning to reach the target states, and (2) Planning Set: VQA tasks that assesses whether MLLM can output correct steps to assemble the LEGO architecture. The experiments show that even the most advanced MLLMs perform far below human levels, which highlights MLLMs deficiencies at multiple-step spatial reasoning.

**Strengths:**

1. The benchmark uses LEGO environment to build a series of novel and interesting tasks.
2. The paper includes a comprehensive tests on latest open-sourced and private models.
3. The benchmark includes both multiple choice questions and QA tasks, which better demonstrate the MLLM capabilities given the model performance quickly decays to 0 as planning steps increasing.

**Weaknesses:**

1. The evaluation does not include the results when MLLMs are trained on these tasks. It could be interesting to demonstrate their performance after trained on these tasks, and especially, if the models trained on spatial understanding tasks could improve their performance on single/multiple step planning.
2. LEGO environment is special and largely different from the realistic environment. Models must correctly understand the spatial information in **virtual** style images, which could pose additional challenges for MLLMs. I appreciate the paper’s effort to measure the correlation coefficients between the proposed benchmark and realistic benchmark and this largely alleviate my concerns, but I would like to highlight this intrinsic limitation of this benchmark.
3. It would be clearer if the paper includes a comparison between LEGO-PUZZLEs with existing benchmark. The paper already includes a related work discussion in section 7, but in a way that listing the current benchmarks. It does not highlight how LEGO-PUZZLEs solve the previous benchmarks problems, such as CLEVR and 3DSRBench.

**Questions:**

Please see the weaknesses above for the three questions about the training results, gap caused by virtual style images, and comparison with existing benchmarks.

---

> ### Author Response · Authors · 2025-11-22
>
> Dear Reviewer B7zR:
>
> Thank you for your detailed and encouraging feedback, which have greatly improved our work. We appreciate your recognition of the benchmark’s **novelty and interesting task design**, its **comprehensive model evaluation**, and its **broad task coverage**. We address your concerns in detail below.
> > ***Weakness 1: Missing evaluation of MLLMs after task-specific training and its effect on planning performance.***
> >
>
> Thank you for your insightful question. We agree that evaluating MLLMs after training on these tasks is worth exploring. To further investigate this, we conducted two additional experiments using Qwen3-VL-8B-Instruct to examine: (1) whether fine-tuning on LEGO-Puzzles spatial understanding tasks improves in-domain performance, and (2) whether such training transfers to single-step and multi-step planning. Below we summarize the results.
>
> 1. **Training on spatial understanding tasks yields clear in-domain improvements.** We fine-tuned Qwen3-VL-8B-Instruct on the Spatial Understanding tasks. The training set and evaluation set were split using an 8:2 ratio. The results are shown below:
>
>
>     | Task | Height | Adjacency | Rotation | Multi-view | Average |
>     | --- | --- | --- | --- | --- | --- |
>     | Before Training | 40.0% | 55.0% | 25.0% | 35.0% | 38.8% |
>     | After Training | 45.0% | 55.0% | 35.0% | 40.0% | 43.8% |
>
>     We observe noticeable gains in *Height*, *Rotation*, and *Multi-view*, suggesting that fine-tuning on the spatial understanding tasks does help the model improve its in-domain performance. This is expected, as training on the same domain naturally enhances the model’s ability to handle similar inputs and task formats.
>
> 2. **However, training on spatial understanding tasks does not reliably transfer to single-step or multi-step planning.**
>
>     We also fine-tuned Qwen3-VL-8B-Instruct on the whole Spatial Understanding tasks and then evaluated it on the Single-Step and Multi-Step Sequential Reasoning tasks . The results are shown below:
>
>     | Task | Dependency | Next Step | Position | Rotation Status | Backwards | Ordering | Outlier | Avg |
>     | --- | --- | --- | --- | --- | --- | --- | --- | --- |
>     | Before Training | 79.0% | 56.0% | 34.0% | 54.0% | 43.0% | 14.0% | 25.0% | 43.6% |
>     | After Training | 77.0% | 56.0% | 38.0% | 49.0% | 46.0% | 11.0% | 27.0% | 43.4% |
>
>     The performance of the single-step and multi-step tasks does not show consistent improvement. Instead, the average score slightly decreases. This suggests that skills learned from the spatial understanding tasks do **not** naturally transfer to long-horizon planning.
>
>
> In conclusion, despite the in-domain improvements observed above, we would still like to emphasize that **LEGO-Puzzles is fundamentally designed as an evaluation benchmark rather than a training dataset**. The dataset is intentionally controlled and relatively compact, focusing on isolating specific reasoning skills rather than providing the diversity and scale required for robust model training. Improving general 3D spatial reasoning and long-horizon planning likely requires much larger and more diverse datasets, covering a wider variety of 3D structures, task formats, and visual configurations.

---

> > ### Author Response · Authors · 2025-11-22
> >
> > > ***Weakness 2*: Concerns on the gap between the virtual LEGO environment and real world.**
> > >
> >
> > Thank you for your insightful question and for acknowledging that “our correlation analysis with a realistic benchmark largely alleviates your concerns.” Although LEGO-Puzzles is based on synthetic LEGO scenes, **the choice of LEGO scenes is deliberate and well-motivated.**
> >
> > 1. **LEGO-Puzzles leverages the LEGO assembly process as a natural environment for evaluating spatial understanding and sequential reasoning.**  The assembly process of a complete LEGO model consists of *discrete, interpretable, and visually consistent* construction steps, each requiring precise understanding of 3D spatial relationships and forming a coherent step-by-step progression. This property allows us to systematically examine spatial reasoning across well-defined transitions, which would be **extremely difficult to achieve with unconstrained natural images.** *This makes LEGO a uniquely effective medium for studying multi-step spatial reasoning in a controlled yet meaningful setting.*
> > 2. **LEGO-Puzzles is grounded in real, physically valid LEGO projects rather than arbitrary synthetic scenes.** All scenes in LEGO-Puzzles are derived from real LEGO projects and official building instructions, ensuring that every object and assembly step corresponds to an authentic, physically feasible structure. As a result, the spatial transformations in our benchmark closely mirror those found in real-world assembly tasks. This realism is further supported by the strong performance correlation with 3DSRBench (*Section 5, Table 4* in paper), which is based on natural images. *These properties collectively mitigate concerns that the benchmark is detached from real-world spatial reasoning.*
> > 3. **LEGO provides a controlled visual environment that keeps the focus on 3D spatial reasoning itself.** We render all steps under identical lighting, camera position, and background so that the only changing factor is the assembled structure itself. This design minimizes potential visual distractions while preserving a clear mapping to real physical assembly processes—unlike natural-image settings where lighting, viewpoint, and background vary unpredictably. *This controlled setup allows us to analyze model performance directly to spatial reasoning ability rather than irrelevant visual artifacts.*
> >
> > In conclusion, LEGO-Puzzles achieves a balance between realism and experimental control, making it an ideal testbed for evaluating both spatial understanding and multi-step sequential reasoning in MLLMs.

---

> ### Author Response · Authors · 2025-11-22
>
> > ***Weakness 3: Missing comparison with existing benchmark***
> >
>
> Thank you for the constructive suggestion. We have added a clearer comparison in *Section 6* of the revised version. Below we explain in detail how LEGO-Puzzles addresses the limitations of the two benchmarks you mentioned—CLEVR and 3DSRBench.
>
> - **CLEVR**. Although both CLEVR and LEGO-Puzzles evaluate visual reasoning, our benchmark solves several key limitations of CLEVR:
>     1. **CLEVR focuses on low-level visual reasoning, while LEGO-Puzzles targets higher-level spatial reasoning and multi-step planning.** CLEVR mainly tests basic abilities such as identifying colors, shapes, counts, and simple spatial relations in a static scene. LEGO-Puzzles goes significantly further by requiring deeper understanding of 3D structure and performing multi-step reasoning across an assembly process. For example, the *PLAN-k-Step* task requires models to identify and order multiple assembly states—a long-horizon planning challenge that CLEVR does not cover.
>     2. **CLEVR uses single-image static questions, while LEGO-Puzzles includes both single-image and multi-image reasoning.** CLEVR evaluates reasoning over one fixed scene. In contrast, LEGO-Puzzles includes tasks based on sequences of images such as predicting the next assembly step, selecting correct intermediate states, and ordering multi-step transitions, thereby capturing dynamic spatial changes that CLEVR cannot model.
>     3. **CLEVR uses abstract, simplified geometric scenes, while LEGO-Puzzles uses realistic and diverse LEGO assembly states.** While CLEVR removes visual complexity to isolate reasoning, LEGO-Puzzles draws from a wide range of real LEGO files, from small (<10 blocks) to large (>100 blocks) structures, with rich color and geometric diversity. This makes LEGO-Puzzles much closer to real-world spatial tasks. In addition, LEGO-Puzzles shows a strong correlation with 3DSRBench (*Section 5, Table 4* in paper), a benchmark based on natural images, demonstrating that the spatial reasoning skills tested by our benchmark generalize beyond toy settings—unlike CLEVR’s highly abstract domain.
> - 3DSRBench: Although both 3DSRBench and LEGO-Puzzles evaluate spatial reasoning, our benchmark addresses several key limitations of 3DSRBench:
>     1. **3DSRBench evaluates static 3D spatial reasoning, while LEGO-Puzzles extends this to multi-step spatial reasoning and planning.** 3DSRBench focuses on understanding 3D relationships such as height, location, orientation and multi-object relations in single images. In contrast, many LEGO-Puzzles tasks require not just interpreting the current state but also imagining how the scene would change after a hypothetical assembly action, and then applying this reasoning across multiple steps. This shift from static perception to multi-step, imagination-based spatial reasoning enables a significantly deeper evaluation of how models handle evolving 3D structures rather than a single snapshot.
>     2. **3DSRBench relies on single-image questions, while LEGO-Puzzles includes both single-image and multi-image reasoning.** Similar to CLEVR, 3DSRBench only evaluayes spatial understanding in a one fixed scene. LEGO-Puzzles introduces sequential tasks that require interpreting changes across multiple images, allowing us to evaluate how models reason about dynamic 3D transitions rather than one fixed scene.

---

> ### Author Response · Authors · 2025-11-27
> **looking forward to your reply**
>
> Dear Reviewer B7zR,
>
> We sincerely appreciate the time, effort, and thoughtful attention you have devoted to reviewing our paper. Your constructive feedback and valuable suggestions have been instrumental in helping us improve our work.
>
> As the author–reviewer discussion period is approaching its end, we wanted to ensure that our responses have fully addressed your concerns, including **additional experiments and analysis addressing the absence of evaluations for task-trained models and clarifying how such training may influence both single- and multi-step planning performance, discussion of the LEGO-Puzzles's generalizability and the rationale behind our design choices, and our strengthened comparison with existing benchmarks.**
>
> If there are still points that you feel need further explanation or discussion, please let us know. We would be more than happy to provide additional details or clarification, and we value the opportunity to address any remaining questions before the discussion phase concludes.
>
> Once again, thank you for your dedication, time, and insights. We deeply value your contribution to the review process and look forward to hearing your thoughts.
>
> Sincerely,
>
> LEGO-Puzzles Authors

---

### Official Review · Reviewer_YBDc · 2025-11-05

**Soundness:** 3
**Presentation:** 3
**Contribution:** 3
**Rating:** 4
**Confidence:** 5

**Summary:**

This paper introduces LEGO-Puzzles, a benchmark designed to assess the multi-step spatial reasoning abilities of multimodal large language models (MLLMs) through LEGO-based visual and planning tasks. Evaluating 23 state-of-the-art models, the study finds that even the strongest systems fall significantly short of human performance, especially in long-horizon planning and spatially grounded image generation, revealing fundamental limitations in current MLLMs’ ability to reason about and plan in 3D space.

**Strengths:**

- High significance: The paper addresses a crucial yet underexplored aspect of multimodal reasoning: multi-step spatial reasoning, which is fundamental to real-world applications such as robotics and embodied AI.

- Systematic and hierarchical benchmark design: LEGO-Puzzles is carefully structured, progressing from basic spatial understanding to single-step and multi-step reasoning, and finally to explicit multi-step planning. This hierarchical organization allows for a comprehensive and fine-grained assessment of spatial intelligence in MLLMs.

- Rich and multi-layered evaluation: Beyond multiple-choice VQA tasks, the paper further evaluates image generation outputs, revealing that reasoning abilities do not readily transfer to generative modalities.

**Weaknesses:**

- **Major overlap with existing work (PhyBlock [1])**: The main concern is the significant conceptual and methodological similarity to the published paper PhyBlock [1]. Both works share nearly identical motivation (evaluating spatial and physical reasoning through block assembly) and evaluation setups (one-step and multi-step prediction). The paper does not cite or discuss PhyBlock, nor does it clarify how LEGO-Puzzles differs in task formulation, data scale, or evaluation scope. A detailed comparison is necessary to establish novelty.

- **Lack of analysis on underlying model mechanisms**: While the experiments clearly expose large performance gaps among MLLMs, the paper does not investigate why these failures occur. There is no analysis of model internals such as visual representation quality, spatial memory, context length limitations, or reasoning strategies that might explain the degradation in multi-step performance.

- **Limited real-world applicability**: Although the authors show a high correlation with 3DSRBench, LEGO-Puzzles is entirely based on rendered synthetic LEGO scenes. This raises concerns about the domain gap between synthetic and real-world images. The benchmark’s external validity for real-world spatial reasoning remains to be empirically demonstrated.

[1] PhyBlock: A Progressive Benchmark for Physical Understanding and Planning via 3D Block Assembly. NeurIPs 25

**Questions:**

Please refer to the Weaknesses part.

---

> ### Author Response · Authors · 2025-11-22
>
> Dear Reviewer YBDc:
>
> Thank you for your detailed and constructive comments, which have greatly helped us improve the paper. We are pleased that you **recognize the significance of our exploration in multi-step spatial reasoning—a crucial yet underexplored aspect of multimodal reasoning, which is fundamental to real-world applications such as robotics and embodied AI.**
>
> > ***[1/2] Weakness1: Concerns on the similarity to PhyBlock; Clarification on the novelty of LEGO-Puzzles.***
> >
>
> We thank the reviewer for raising this important question. Below we address your concerns in detail:
>
> These two works were developed **concurrently**, and we have added an explicit discussion and citation in the revised version (*Section 6*). Although both benchmarks evaluate the spatial reasoning capabilities of MLLMs through block assembly, the most significant difference lies in their core focus, task formulation, and evaluation principles, among others. PhyBlock centers on *physical* *understanding* such as object stability and spatial support, while LEGO-Puzzles is explicitly designed to evaluate ***multi-step reasoning*** in MLLMs through controlled and parameterized planning horizons. These distinct focuses lead to fundamentally different methodological choices and evaluation principles. Below we provide a detailed comparison to clarify the novelty of LEGO-Puzzles.
>
> 1. **LEGO-Puzzles provides explicit and fine-grained control over multi-step planning difficulty.**
>     - PhyBlock organizes its planning tasks into 4 “levels,” where higher levels **roughly** correspond to more blocks. However, the planning horizon is not an explicit parameter but is instead implicitly tied to the number of blocks in the structure. As a result, “multi-step difficulty“ cannot be precisely aligned across tasks. This loose definition is also reflected in their reported results: 4 out of 7 open-source models achieve higher recall score on Level 4 than on Level 3, indicating that this **coarse, block-count–based grouping does not provide a reliable measure of planning complexity.**
>     - In contrast, **LEGO-Puzzles directly parameterizes difficulty through the step count**. Our Planning Set defines a clear task: PLAN-k-Step (k = 1…8), requiring the model to plan from $x_0$ to $x_{k+1}$ through exactly k intermediate states, each explicitly provided and evaluated. This design produces a clean and controlled benchmark for assessing multi-step planning. Experiments in *Section 3.2.2* further validate this parameterization—performance degrades smoothly and consistently as k increases.
> 2. **LEGO-Puzzles defines a harder planning task, with stricter and more realistic evaluation.** Unlike PhyBlock, **LEGO-Puzzles can scale planning difficulty arbitrarily**, and our experiments show that at 8-step planning, all models drop to near 0 accuracy (benchmark saturation). This saturation does not occur in PhyBlock even at its highest difficulty (Level 4), where all models still maintain relatively high recall. We believe this gap comes from two key factors:
>     1. **LEGO-Puzzles uses more complex, more realistic, and more confusing distractors.** Our negative candidates differ from the ground-truth states only by subtle but realistic perturbations—e.g., slight color mismatches, minor misplacement, or near-correct partial assemblies. These distractors simulate noisy real-world planning conditions, and force fine-grained spatial discrimination. PhyBlock’s blocks, on the other hand, are **heavily abstracted**: only (1). **8 block types**, (2). **5 colors**, and (3). maximum **22 blocks per scene**, resulting in a scene that is overly abstract and far simpler than real-world assemblies.
>     2. **The evaluation protocol in LEGO-Puzzles is designed to better reflect real-world planning requirements.** It requires the model to provide a fully correct k-step plan to receive any credit, so the model must get the entire sequence right. This design also naturally captures the real-world phenomenon that errors accumulate as the number of planning steps increases. In contrast, PhyBlock scores each step independently and gives partial credit even when the full plan is wrong, which makes the task easier and **less sensitive to long-horizon mistakes.**

---

> ### Author Response · Authors · 2025-11-22
>
> > ***[2/2] Weakness 1: Concerns on the similarity to PhyBlock; Clarification on the novelty of LEGO-Puzzles.***
> >
>
> 3. **LEGO-Puzzles employs richer, more realistic block structures that better reflect real-world spatial complexity.**
>     - PhyBlock states that its blocks are “curated from the Internet for inspiration,” but the final objects remain highly simplified—only **8 shapes**, **5 colors**, and at most **22 pieces** per scene. In many of their Level 3 and Level 4 examples, we also observe that **multiple blocks are exactly identical**, leading to very low structural diversity and a toy-like appearance.
>     - In contrast, **LEGO-Puzzles uses diverse, open-source LEGO files** spanning: very small builds (<10 pieces), medium structures, and large, complex assemblies (**>100 pieces**), with **more than 30 colors and over 100 different brick shapes**. Furthermore, LEGO-Puzzles correlates strongly with 3DSRBench (*Section 5, Table 4* in paper), a real-world spatial reasoning benchmark. This indicates that the abilities tested in our benchmark generalize beyond toy block assembly. By comparison, PhyBlock does not report any such empirical correlation with real-world data, leaving its connection to practical scenarios less substantiated.
> 4. **LEGO-Puzzles and PhyBlock are concurrent works.** We would like to note that PhyBlock was accepted by *NeurIPS 2025 in September*, which falls within the **four-month** window defined by the ICLR contemporaneous-work policy. This policy states as follows: “We consider papers contemporaneous if they are published within the last four months. That means, since our full paper deadline is October 1, if a paper was published (i.e., at a peer-reviewed venue) on or after July 1, 2024, authors are not required to compare their own work to that paper. Authors are encouraged to cite and discuss all relevant papers, but they may be excused for not knowing about papers not published in peer-reviewed conference proceedings or journals, which includes papers exclusively available on arXiv”. We would like to thank the reviewer for pointing out the PhyBlock paper. In the revised version, we have now added an explicit citation and a discussion in *Section 6* to clarify the relationship and differences between these two benchmarks.
>
> > ***[1/2] Weakness 2:** **Lack of analysis on underlying model mechanisms***
> >
>
> We thank the reviewer for this insightful suggestion. Below we provide additional analysis in detail:
>
> 1. **Limitations in forming stable 3D spatial representations.** Findings from *Section 3.2.1* of the paper provide additional evidence that current MLLMs lack a reliable 3D spatial understanding. In the *Height* task, even when the prompt explicitly specifies a 3D environment, many models produce answers that are consistent with what one would obtain using only a 2D perspective. This pattern suggests that the models often interpret the scene from a 2D perspective rather than a 3D perspective. In the *Rotation* task, many models tend to output the same rotation angle even when the input indicates different rotation degrees. This shows that they fail to reliably perceive and distinguish changes in object orientation, suggesting that the models struggle to encode and differentiate 3D rotational transformations. These limitations likely propagate into multi-step planning: without a stable 3D understanding of intermediate states, the model cannot maintain coherent spatial information across steps, leading to compounding errors in Plan-k-Step as the sequence becomes longer.

---

> > ### Author Response · Authors · 2025-11-22
> >
> > > ***[2/2] Weakness 2:** **Lack of analysis on underlying model mechanisms***
> > >
> >
> > 2. **The failures of current MLLMs on multi-step sequential reasoning stem from genuine capability limitations rather than prompt-format or domain-mismatch issues.**
> >
> >     To further investigate the model’s internal mechanisms, **we conducted two additional experiments** using Qwen3-VL-8B-Instruct to examine: (1) whether fine-tuning on LEGO-Puzzles spatial understanding tasks improves in-domain performance, and (2) whether such training transfers to single-step and multi-step planning. Below we summarize the results:
> >
> >     - **Training on spatial understanding tasks yields clear in-domain improvements.** We fine-tuned Qwen3-VL-8B-Instruct on the Spatial Understanding tasks. The training set and evaluation set were split using an 8:2 ratio. The results are shown below:
> >
> >
> >         | Task | Height | Adjacency | Rotation | Multi-view | Average |
> >         | --- | --- | --- | --- | --- | --- |
> >         | Before Training | 40.0% | 55.0% | 25.0% | 35.0% | 38.8% |
> >         | After Training | 45.0% | 55.0% | 35.0% | 40.0% | 43.8% |
> >
> >         We observe noticeable gains in *Height*, *Rotation*, and *Multi-view*, suggesting that fine-tuning on the spatial understanding tasks does help the model improve its in-domain performance. This is expected, as training on the same domain naturally enhances the model’s ability to handle similar inputs and task formats.
> >
> >     - **However, training on spatial understanding tasks does not reliably transfer to single-step or multi-step planning.**
> >
> >         We also fine-tuned Qwen3-VL-8B-Instruct on the whole Spatial Understanding tasks and then evaluated it on the Single-Step and Multi-Step Sequential Reasoning tasks. The results are shown below:
> >
> >         | Task | Dependency | Next Step | Position | Rotation Status | Backwards | Ordering | Outlier | Avg |
> >         | --- | --- | --- | --- | --- | --- | --- | --- | --- |
> >         | Before Training | 79.0% | 56.0% | 34.0% | 54.0% | 43.0% | 14.0% | 25.0% | 43.6% |
> >         | After Training | 77.0% | 56.0% | 38.0% | 49.0% | 46.0% | 11.0% | 27.0% | 43.4% |
> >
> >         The performance of the single-step and multi-step tasks does not show consistent improvement. Instead, the average score slightly decreases. This suggests that skills learned from the spatial understanding tasks do not naturally transfer to long-horizon planning.
> >
> >
> >     Based on the experiments above, our analysis is as follows:
> >
> >     - Although fine-tuning on the Spatial Understanding tasks yields *some* in-domain gains, the improvements remain modest. This indicates that models can learn localized spatial cues when the task structure and input distribution closely align with training data, but the extent of improvement is limited.
> >     - These in-domain gains do not transfer to either single-step or multi-step planning: performance on sequential reasoning tasks stays essentially unchanged or even slightly decreases after training. This shows that the failure of multi-step planning is not due to prompt-format issues or domain mismatch.
> >     - Together, these observations reveal that current MLLMs lack the underlying mechanisms needed for long-horizon spatial reasoning—such as maintaining spatial state across steps, updating intermediate configurations, and integrating 3D structure over time. The observed degradation therefore reflects fundamental representational and architectural limitations.
> >
> > In conclusion, current MLLMs still have substantial room for improvement in multi-step sequential reasoning. At the same time, diagnosing the internal failure mechanisms of large neural networks is  in general a challenging open problem. We view our experiments as a first step toward revealing *what* fails behaviorally. If the reviewer has particular hypotheses or analysis directions, we would be very happy to incorporate additional experiments.

---

> ### Author Response · Authors · 2025-11-22
>
> > ***Weakness 3: Concerns on limited real-world applicability***
> >
>
> Thank you for raising this important point. Although LEGO-Puzzles is based on synthetic LEGO scenes, **the choice of LEGO scenes is deliberate and well-motivated.** Below, we address each aspect in detail:
>
> 1. **LEGO-Puzzles leverages the LEGO assembly process as a natural environment for evaluating spatial understanding and sequential reasoning.** The assembly process of a complete LEGO model consists of *discrete, interpretable, and visually consistent* construction steps, each requiring a precise understanding of 3D spatial relationships and forming a coherent step-by-step progression. This property allows us to systematically examine spatial reasoning across well-defined transitions, which would be **extremely difficult to achieve with unconstrained natural images.** *This makes LEGO a uniquely effective medium for studying multi-step spatial reasoning in a controlled yet meaningful setting.*
> 2. **LEGO-Puzzles is grounded in real, physically valid LEGO projects rather than arbitrary synthetic scenes.** All scenes in LEGO-Puzzles are derived from real LEGO projects and official building instructions, ensuring that every object and assembly step corresponds to an authentic, physically feasible structure. As a result, the spatial transformations in our benchmark closely mirror those found in real-world assembly tasks. This realism is further supported by the strong performance correlation with 3DSRBench (*Section 5, Table 4 in the paper*), which is based on natural images. *These properties collectively mitigate concerns that the benchmark is detached from real-world spatial reasoning.*
> 3. **LEGO provides a controlled visual environment that keeps the focus on 3D spatial reasoning itself.** We render all steps under identical lighting, camera position, and background so that the only changing factor is the assembled structure itself. This design minimizes potential visual distractions while preserving a clear mapping to real physical assembly processes—unlike natural-image settings where lighting, viewpoint, and background vary unpredictably. *This controlled setup allows us to analyze model performance directly to spatial reasoning ability rather than irrelevant visual artifacts.*
>
> In conclusion, LEGO-Puzzles achieves a balance between realism and experimental control, making it an ideal testbed for evaluating both spatial understanding and multi-step sequential reasoning in MLLMs.

---

> ### Author Response · Authors · 2025-11-27
> **looking forward to your reply**
>
> Dear Reviewer YBDc,
>
> We sincerely appreciate the time, effort, and thoughtful attention you have devoted to reviewing our paper. Your constructive feedback and valuable suggestions have been instrumental in helping us improve our work.
>
> As the author–reviewer discussion period is approaching its end, we wanted to ensure that our responses have fully addressed your concerns, including **the comparison with PhyBlock and the clarification of LEGO-Puzzles’ novelty, the expanded analysis of underlying model mechanisms that may explain the performance gaps in multi-step reasoning, as well as our discussion of the benchmark’s generalizability and the rationale behind our design choices.**
>
> If there are still points that you feel need further explanation or discussion, please let us know. We would be more than happy to provide additional details or clarification, and we value the opportunity to address any remaining questions before the discussion phase concludes.
>
> Once again, thank you for your dedication, time, and insights. We deeply value your contribution to the review process and look forward to hearing your thoughts.
>
> Sincerely,
>
> LEGO-Puzzles Authors

---

### Author Response · Authors · 2025-12-03
**General Response [1/2]**

Thank you and all the reviewers for your time and the constructive feedback on our submission. We have carefully considered every comment and prepared a detailed rebuttal. Below we summarize (1) the strengths highlighted by the reviewers, (2) the key concerns and how we addressed them, and (3) the reviewers’ final positions. We also note an important update: **Reviewer 2tpf explicitly stated an intention to increase the rating after our responses resolved the major concerns.**

### **1. Strengths Highlighted by the Reviewers**

We appreciate the strong support from all reviewers (YBDc, B7zR, 2tpf, z8xn), who consistently highlighted several core strengths of LEGO-Puzzles:

- **Clear contribution and high significance.** LEGO-Puzzles fills an important gap by evaluating *multi-step* and *spatially grounded* reasoning—an ability fundamental to real-world applications such as robotics and embodied AI, yet largely underexplored in existing benchmarks.(Reviewers YBDc, B7zR, 2tpf, z8xn)
- **Comprehensive and transparent benchmark design.** LEGO-Puzzles provides a well-structured hierarchical task design, a detailed and reproducible data curation pipeline, and clear presentation of evaluation settings. (Reviewers YBDc, 2tpf, z8xn)
- **Extensive empirical evaluation.** The benchmark provides a comprehensive evaluation of 23 state-of-the-art MLLMs, spanning both open-source and proprietary models, and offers convincing landscape of their current limitations. (Reviewers YBDc, B7zR, 2tpf, z8xn)

### **2. Concerns and How We Addressed Them**

> **1. Concerns on the similarity to PhyBlock; clarification on the novelty of LEGO-Puzzles.** (Reviewer YBDc)
>

We provided a detailed comparison and clarified that LEGO-Puzzles differs fundamentally in its (1) focus (multi-step planning vs. physical stability), (2) explicit difficulty control through PLAN-k-Step, (3) stricter and more realistic evaluation, and (4) substantially higher structural diversity. We also clarified that the two works are concurrent under ICLR’s policy and added citation and discussion accordingly.

> **2. Lack of analysis on underlying model mechanisms.** (Reviewer YBDc)
>

We added new fine-tuning experiments and analyses showing that current models lack stable 3D spatial representations, and that improvements on basic spatial tasks do not transfer to multi-step planning, indicating that failures stem from fundamental capability limitations rather than prompt or domain issues.

> **3. Concerns on the gap between the virtual LEGO environment and the real world.** (Reviewer YBDc, B7zR, z8xn)
>

We clarified that the choice of LEGO scenes is deliberate and well-motivated: (1) the LEGO assembly process provides discrete and physically meaningful 3D transitions difficult to obtain from natural images; (2) all scenes come from real, physically valid LEGO projects, and their strong correlation with 3DSRBench supports real-world relevance; and (3) the controlled rendering setup isolates spatial reasoning by removing irrelevant visual variation.

> **4. Missing evaluation of MLLMs after task-specific training and its effect on planning performance.** (Reviewer B7zR)
>

We added new fine-tuning experiments on the spatial-understanding tasks and found that although this yields small in-domain gains, it does not transfer to single- or multi-step planning, indicating that planning failures arise from inherent model limitations.

> **5. Missing comparison with existing benchmarks: CLEVR and 3DSRBench.** (Reviewer B7zR)
>

We added a clearer comparison in the revision, showing how LEGO-Puzzles addresses the limitations of existing benchmarks.

> **6. Concerns on simplifying planning to a discriminative task instead of true generative planning.** (Reviewer 2tpf)
>

We added new generative-planning experiments and found that even frontier models fail completely (0% success), confirming that full generative planning is far beyond current capabilities. This validates the discriminative PLAN-k-Step formulation as an appropriate and informative task setting.

> **7. Questions regarding the inconsistency of human performance across different tasks.** (Reviewer 2tpf)
>

We clarified that the difference is expected: the Height task in the Elementary set contains intentional 2D–3D ambiguities that can lead to occasional human mistakes (e.g., relying on quick visual intuition when blocks are far apart). We also found that these errors mainly come from not inspecting the images carefully; when participants were explicitly reminded to pay closer attention, they reached 100% accuracy (see examples in Appendix 8.2). In contrast, the Planning tasks do not require such fine height estimation—step differences are usually clearer (e.g., added pieces, color changes, or part movements). As a result, humans perform perfectly on the Planning tasks.

---

### Author Response · Authors · 2025-12-03
**General Response [2/2]**

> **8. Evaluation in agentic/interactive environments and structuring the task as an RLVR environment.** (Reviewer 2tpf)
>

We clarified that LEGO-Puzzles is designed as an MLLM benchmark for foundational multi-step spatial reasoning, and applying it to agentic settings or structuring it as an RLVR environment would require a full interactive simulation stack—beyond the current scope. Nevertheless, the step-by-step, verifiable ground truth makes it a natural foundation for future agentic extensions.

> **9. Prompting strategies for mitigating 2D projection bias in 3D tasks.** (Reviewer 2tpf)
>

We added experiments with explicit 3D instructions, CoT, and few-shot prompts and found that none meaningfully mitigate the 2D bias; improvements are small and inconsistent, indicating that the limitation stems from model capabilities rather than prompting.

> **10. Evaluation of differences between multimodal reasoning models and chat models.** (Reviewer 2tpf)
>

We compared a chat-oriented model (Qwen3-VL-8B-Instruct) with its reasoning-enhanced counterpart (Qwen3-VL-8B-Thinking) and found no performance advantage for the reasoning model. This shows that stronger reasoning fine-tuning does not address core limitations in multimodal spatial reasoning.

> **11. Concerns on limited reproducibility due to human-scored image generation; comparison with LLM-as-a-judge.** (Reviewer z8xn)
>

We added a comparison between human scoring and LLM-as-a-judge evaluation and found substantial disagreement—low or even negative correlations, showing that current LLM judges are not yet reliable for spatially precise image-generation tasks. Human scoring remains necessary.

> **12. Lack of few-shot and prompt-sensitivity analysis.** (Reviewer z8xn)
>

We added few-shot and prompt-sensitivity experiments showing that few-shot prompting offers only modest improvements and that models are largely insensitive to image placement. These results confirm that core limitations persist regardless of prompting strategy.

> **13. Need tests with composite images or varying image counts.** (Reviewer z8xn)
>

We added tests using composite images and found that performance drops when more than six images are merged into a single overloaded composite, confirming that LEGO-Puzzles’ multi-image input design is reasonable.

## **3. Reviewers’ Final Positions**

- Reviewer z8xn: Rating: 8
- Reviewer B7zR: Rating: 6
- Reviewer YBDc: Rating: 4; we have tried to thoroughly address all of the review’s concerns, but received no reply before further reviewer discussions or public comments were closed.
- Reviewer 2tpf: Initially rated 4, but explicitly stated: ***“since the authors have resolved the majority of my concerns, I intend to increase my rating,”*** indicating an updated score of at least 6.

In conclusion, all reviewers agree that LEGO-Puzzles evaluates an important and underexplored capability in MLLMs, filling a clear gap in assessing multi-step and spatially grounded reasoning. Throughout the rebuttal and discussion, we provided extensive new experiments, detailed comparisons, and clarified the design rationale. No major technical concerns remain unresolved, and **one reviewer explicitly stated an intention to increase the rating.**

We sincerely hope you will take this context into account when assessing our paper. Thank you again for your time and consideration.

---

### Meta-Review · Area_Chair_pzZ3 · 2025-12-31

**Summary:**

This paper introduces a new benchmark, namely LEGO Puzzle. It tests how well MLLMs can reason and plan in 3D space. It is structured into two primary components: the Elementary Set, which consists of 11 VQA tasks (approx. 1,100 samples) covering basic spatial understanding, adjacency, and sequential logic. Planning Set (PLAN-k-Step): Requires models to generate step-by-step assembly instructions (from 1 to 8 steps) to reach a target configuration. The authors tested 23 existing VLMs, including both open-sourced and private ones, on the proposed benchmarks. The study reveals fundamental deficiencies in MLLMs' 3D spatial reasoning, as even top-tier models significantly underperform compared to humans in long-horizon planning and grounded image generation.

While the reviewers and AC recognized that introducing sequential planning into a 3D LEGO-based benchmark—building upon the foundations of PhyBlock, CLEVR, and 3DSRBench—significant concerns remain regarding its external validity. Specifically, three of the reviewers are skeptical about whether the failure cases observed in a synthetic LEGO environment are truly representative of model limitations in the real world, which limits the benchmark's broader impact on the field. Additionally, the AC also concerns **the size of the benchmark**.

**Reviewer Concerns:**

After reviewing all the reviews and rebuttals, some reviews are addressed, but some shared concerns still remain.

**Novelty and Concurrent Work**
Reviewer YBDc noted a significant overlap with PhyBlock (NeurIPS 2025). The authors argue that this is "concurrent work" per ICLR policy. While this may excuse them from a formal baseline comparison in the initial submission, the conceptual delta between the two works remains thin. The authors’ claim that LEGO-Puzzles offers more "controlled parameterization" is noted, but the fundamental task remains similar to existing benchmarks.

Meanwhile, the reviewer raised a concern that the authors did not **explore the underlying mechanism**, and such 3D limitations in the LEGO environment cannot generalize to real scenarios. The other two reviewers (z8xn and B7zR) also share the same concern. To further validate the reviewer's concerns, the authors conducted fine-tuning on the proposed benchmark pairs.
 1. Fine-tuning an 8-billion-parameter model on only ~880 samples (80% of the 1,100 pairs) is statistically risky. This volume of data is negligible compared to the model's pre-training corpus. Such a small signal is more likely to cause overfitting to the prompt template or catastrophic forgetting rather than a genuine shift in spatial reasoning capabilities.
2. The authors report a 5% average improvement in "Spatial Understanding" after training. On their tiny evaluation set of 220 pairs, a 5% shift represents a difference of only ~11 questions. In the context of LLM evaluation, this is within the margin of stochastic noise and does not provide robust evidence that the model "learned" 3D spatial representations.
3. The authors argue that the lack of transfer from spatial understanding to planning proves a "fundamental architectural limitation." However, it is equally likely that the model simply failed to learn anything meaningful from the sparse training data provided. Consequently, the experiment does not successfully "diagnose" the internal failure as claimed.

Some minor constructive concerns, such as "Evaluation in agentic/interactive environments and structuring the task as an RLVR environment" and " limited reproducibility due to human-scored image generation; comparison with LLM-as-a-judge" are still open to discuss.

**Reviewer Scores:**

Original ratings from the reviewers are:
Reviewer z8xn: 8
Reviewer B7zR: 6
Reviewer YBDc: 4
Reviewer 2tpf: 4

Although reviewer 2tpf is willing to increase the score, his/her major concerns do not overlap with the major concerns from the other three reviewers.

---

### Decision · Program_Chairs · 2026-01-26

Reject